# HYBRID MILP TO EFFICIENTLY AND ACCURATLY SOLVE HARD DNN VERIFICATION INSTANCES

## ABSTRACT

$\alpha, \beta$-CROWN has won the last 4 VNNcomp(etitions), as the DNN verifier with the best trade-off between accuracy and runtime. VNNcomp however is focusing on relatively easy verification instances. In this paper, we consider *harder* verification instances, on which $\alpha, \beta$-Crown displays a large number ($20 - 58\%$) of undecided instances, that is, instances that can neither be verified, nor an explicit attack can be found. Enabling larger time-outs for $\alpha, \beta$-Crown only improves verification rate by few percents, leaving a large gap of undecided instances while already taking a considerable amount of time. Resorting to slow complete verifiers, does not fare better even with very large time-outs: They would theoretically be able to close the gap, but with an untractable runtime on all but small *hard* instances.

In this paper, we propose a novel Utility function that selects few neurons to be encoded with accurate but costly integer variables in a *partial MILP* problem. The novelty resides in the use of the solution of *one* (efficient LP) solver to accurately compute a selection $\varepsilon$-optimal for a given input. Compared with previous attempts, we can reduce the number of integer variables by around 4 times while maintaining the same level of accuracy. Implemented in *Hybrid MILP*, calling first $\alpha, \beta$-Crown with a short time-out to solve easier instances, and then partial MILP for those for which $\alpha, \beta$-Crown fails, produces a very accurate yet efficient verifier, reducing tremendously the number of undecided instances ($8 - 15\%$), while keeping a reasonable runtime ($46s - 417s$ on average per instance).

## 1 INTRODUCTION

Deep neural networks (DNNs for short) have demonstrated remarkable capabilities, achieving human-like or even superior performance across a wide range of tasks. However, their robustness is often compromised by their susceptibility to input perturbations Szegedy et al. (2014). This vulnerability has catalyzed the verification community to develop various methodologies, each presenting a unique balance between completeness and computational efficiency Katz et al. (2019; 2017); Singh et al. (2019b). This surge in innovation has also led to the inception of competitions such as VN-NComp Brix et al. (2023b), which aim to systematically evaluate the performance of neural network verification tools. While the verification engines are generic, the benchmarks usually focus on local robustness, i.e. given a DNN, an image and a small neighbourhood around this image, is it the case that all the images in the neighbourhood are classified in the same way. For the past 5 years, VNNcomp has focused on rather easy instances, that can be solved within tens of seconds (the typical hard time-out is 300s). For this reason, DNN verifiers in the past years have mainly focused on optimizing for such easy instances. Among them, NNenum Bak (2021), Marabou Katz et al. (2019); Wu et al. (2024), and PyRAT Durand et al. (2022), respectively 4th, 3rd and 2sd of the last VNNcomp'24 Brix et al. (2024) and 5th, 2sd and 3rd of the VNNcomp'23 Brix et al. (2023a); Mn-BAB Ferrari et al. (2022), 2sd in VNNcomp'22 Müller et al. (2022), built upon ERAN Singh et al. (2019b) and PRIMA Müller et al. (2022); and importantly, $\alpha, \beta$-Crown Wang et al. (2021); Xu et al. (2021), the winner of the last 4 VNNcomp, benefiting from branch-and-bound based methodology Zhang et al. (2022); Bunel et al. (2020). We will thus focus in the following mostly on $\alpha, \beta$-Crown.

Easy instances does not mean small DNNs: for instance, a ResNet architecture for CIFAR10 (with tens of thousands of neurons) has been fully checked by $\alpha, \beta$-Crown Wang et al. (2021), each instance taking only a couple of seconds to either certify that there is no robustness attack, or finding a

| Network Perturbation | Accuracy | Upper Bound | $\alpha, \beta$-Crown TO=10s | $\alpha, \beta$-Crown TO=30s | $\alpha, \beta$-Crown TO=2000s |
|---|---|---|---|---|---|
| MNIST $5\times100$ $\epsilon = 0.026$ | 99% | 90% | 33% 6.9s | 35% 18.9s | 40% 1026s |
| MNIST $5\times200$ $\epsilon = 0.015$ | 99% | 96% | 46% 6.5s | 49% 16.6s | 50% 930s |
| MNIST $8\times100$ $\epsilon = 0.026$ | 97% | 86% | 23% 7.2s | 28% 20.1s | 28% 930s |
| MNIST $8\times200$ $\epsilon = 0.015$ | 97% | 91% | 35% 6.8s | 36% 18.2s | 37% 1083s |
| MNIST $6\times500$ $\epsilon = 0.035$ | 100% | 94% | 41% 6.4s | 43% 16.4s | 44% 1003s |
| CIFAR CNN-B-adv $\epsilon = 2/255$ | 78% | 62% | 34% 4.3s | 40% 8.7s | 42% 373s |
| CIFAR ResNet $\epsilon = 2/255$ | 29% | 25% | 25% 2s | 25% 2s | 25% 2s |

Table 1: Images verified by $\alpha, \beta$-Crown with different time-outs (TO) on 7 DNNs, and average runtime per image. The 6 first DNNs are hard instances. The last DNN (ResNet) is an easy instance (trained using Wong to be easy to verify, but with a very low accuracy level), provided for reference.

very close neighbour with a different decision. One issue is however that easy instances are trained specifically to be easier to verify e.g. using DiffAI Mirman et al. (2018) PGD Madry et al. (2018), which can impact the accuracy of the network, i.e. answering correctly to an unperturbed input. For instance, this ResNet was trained using Wong, and only 29% of its answers are correct Müller et al. (2022) (the other 71% are thus not tested). While more accurate trainers for verification have been recently developed Xu et al. (2024), they can only simplify one given verification specification by a limited amount before hurting accuracy, turning e.g. very hard verification instances into hard verification instances. Also, verification questions intrinsically harder than local robustness, such as bounding on Lipschitz constants Wang et al. (2022) globally or asking several specification at once, makes the instance particularly harder. Last, there are many situations (workflow, no access to the dataset...) where using specific trainers to learn easy to verify DNN is simply not possible, leading to *verification-agnostic* networks Dathathri et al. (2020). The bottom line is, one cannot expect only *easy* verification instances: *hard* verification instances need to be explored as well.

In this paper, we focused on the 6 *hard* ReLU-DNNs that have been previously tested in Wang et al. (2021), which display a large gap ($\geq 20\%$) between images that can be certified by $\alpha, \beta$-Crown and the upper bound when we remove those which can be falsified. In turns, hard instances does not necessarily mean very large DNNs, the smallest of these hard DNNs having only 500 hidden neurons, namely MNIST $5\times100$. We first dwelve into the scaling of $\alpha, \beta$-Crown, to understand how longer Time-Out (TO) affects the number of undecided images and the runtime. Table 1 reveals that even allowing for 200 times longer time outs only improves the verification from 2% to 8%, leaving a considerable $20\% - 50\%$ gap of undecided images, while necessitating vastly longer runtime (300s-1000s in average per instance).

The size of the smallest DNN (500 hidden neurons) makes it believable to be solved by complete verifiers such as Marabou 2.0, NNenum or a Full MILP encoding. While they should theoretically be able to close the gap of undecided images, in practice, even with a large 10 000s Time-out, Table 2 reveals that only NNenum succeeds to verify images not verified by $\alpha, \beta$-Crown, limited to 9% more images out of the 50% undecided images, and with a very large runtime of almost 5000s per image. It seemed pointless to test complete verifiers on larger networks.

| Network | Accuracy | Upper | Marabou 2.0 | NNenum | Full MILP |
|---|---|---|---|---|---|
| MNIST $5\times100$ $\epsilon = 0.026$ | 99% | 90% | 28% 6200s | 49% 4995s | 40 % 6100s |

Table 2: Result of complete verifiers on the hard 5x100 with TO = 10 000s. Complete verifier barely (9% out of 50%) outperform $\alpha, \beta$-Crown (40%, 1026s), despite much larger runtime.

Our main contributions address the challenges of verifying *hard* DNNs efficiently:

1. We designed a novel Utility function to choose few neurons to encode with the exact MILP encoding, while others are treated with the efficient LP relaxation, giving rise to partial MILP (pMILP). Specifically, the novelty of Utility resides in the use of the solution to an (efficient LP) solver on the node $z$ we want to bound. Utility can then precisely evaluate how much accuracy is gained by switching neuron $a$ from LP (solution of the LP call) to the exact MILP encoding of ReLU (exact computation from the solution, which can be made thanks to Proposition 1), with a proved bound on the precision (Proposition 2). Because pMILP focuses on the *improvement* (binary - linear), it is much more efficient ($\approx 4$ times less integer variables for same accuracy (Table 6)) than previous attempts, which consider the generic *sensitivity* to this neuron. To the best of our knowledge, this is the first time such a solution of an (LP) call is used to evaluate the contribution of each neuron, including heuristics for BaB, e.g. Bunel et al. (2020); De Palma et al. (2021).

2. We then propose a new verifier, called *Hybrid MILP*, invoking first $\alpha, \beta$-Crown with short time-out to settle the easy instances. On those (*hard*) instances which are neither certified nor falsified, we call pMILP with few neurons encoded as integer variables. Experimental evaluation reveals that Hybrid MILP achieves a beneficial balance between accuracy and completeness compared to prevailing methods. It reduces the proportion of undecided inputs from $20-58\%$ ($\alpha, \beta$-Crown with 2000s TO) to $8-15\%$, while taking a reasonable average time per instance ($46-420$s), Table 3. It scales to fairly large networks such as CIFAR-10 CNN-B-Adv Dathathri et al. (2020), with more than 20 000 neurons.

Limitation: We consider DNNs employing the standard ReLU activation function, though our findings should extend to other activation functions, following similar extention by Huang et al. (2020).

## 1.1 RELATED WORK

We compare Hybrid MILP with major verification tools for DNNs to clarify our methodology and its distinction from the existing state-of-the-art. It scales while preserving good accuracy, through targeting a limited number of binary variables, stricking a good balance between exact encoding of a DNN using MILP Tjeng et al. (2019) (too slow) and LP relaxation (too inaccurate). MIP-planet Ehlers (2017b) opts for a different selection of binary variables, and execute one large MILP encoding instead of Hybrid MILP's many small encodings, which significantly reduce the number of binary variables necessary for each encoding. In Huang et al. (2020), small encodings are also considered, however with a straightforward choice of binary nodes based on the weight of outgoing edges, which need much more integer variables (thus runtime) to reach the same accuracy.

Hybrid MILP can be seen as a refinement of $\alpha, \beta$-Crown Wang et al. (2021), though its refined accurate path is vastly different than the base Branch and Bound technique used in $\alpha, \beta$ CROWN, BaBSR Bunel et al. (2020) and MN-BaB Ferrari et al. (2022), which call BaB once per output neuron. In the worst case, this involves considering all possible ReLU configurations, though branch and bound typically circumvents most possibilities. In simple networks, like those trained robustly, branch and bound is highly efficient, focusing on branches crucial for verifying the actual property. However, branch and bound hits a complexity barrier when verifying harder instances, due to an overwhelming number of branches, as displayed in Table 1. This is not the case of Hybrid MILP, see Table 3, which is much more accurate than $\alpha, \beta$-Crown. That shortcoming for hard instances was witnessed in Wang et al. (2021), and a very specific solution using the full MILP encoding for the first few layers of a DNN was drafted, following similar proposal Singh et al. (2019c). The main issue is that it is slow and it cannot scale to DNNs with many neurons, as every neurons are encoded using an integer variable, making it not that accurate for intermediate networks (e.g. $9 \times 100$, $9 \times 200$, Table 3), and not usable for larger DNNs ($6 \times 500$, CNN-B-Adv), whereas Hybrid MILP does scale.

Last, ERAN-DeepPoly Singh et al. (2019b) computes bounds on values very quickly, by abstracting the weight of every node using two functions: an upper function and a lower function. While the upper function is fixed, the lower function offers two choices. It relates to the LP encoding through the following new (to our knowledge) insight: Proposition 1 state that the LP relaxation precisely matches the intersection of these two choices. Consequently, LP is more accurate (but slower) than DeepPoly, and Hybrid MILP is considerably more precise. Regarding PRIMA Müller et al. (2022), the approach involves explicitly maintaining dependencies between neurons.

Finally, methods such as Reluplex / Marabou Katz et al. (2017; 2019) abstract the network: they diverge significantly from those abstracting values such as PRIMA, $\alpha, \beta$-CROWN)Müller et al. (2022); Wang et al. (2021), Hybrid MILP. These network-abstraction algorithms are designed to be *complete* but completeness comes at the price of significant scalability challenges, and in practice they time-out on hard instances as shown in Table 2.

## 2 NOTATIONS AND PRELIMINARIES

In this paper, we will use lower case latin $a$ for scalars, bold $\boldsymbol{z}$ for vectors, capitalized bold $\boldsymbol{W}$ for matrices, similar to notations in Wang et al. (2021). To simplify the notations, we restrict the presentation to feed-forward, fully connected ReLU Deep Neural Networks (DNN for short), where the ReLU function is $ReLU : \mathbb{R} \to \mathbb{R}$ with $ReLU(x) = x$ for $x \geq 0$ and $ReLU(x) = 0$ for $x \leq 0$, which we extend componentwise on vectors.

An $\ell$-layer DNN is provided by $\ell$ weight matrices $\boldsymbol{W}^i \in \mathbb{R}^{d_i \times d_{i-1}}$ and $\ell$ bias vectors $\boldsymbol{b}^i \in \mathbb{R}^{d_i}$, for $i = 1, \ldots, \ell$. We call $d_i$ the number of neurons of hidden layer $i \in \{1, \ldots, \ell - 1\}$, $d_0$ the input dimension, and $d_\ell$ the output dimension.

Given an input vector $\boldsymbol{z}^0 \in \mathbb{R}^{d_0}$, denoting $\hat{\boldsymbol{z}}^0 = \boldsymbol{z}^0$, we define inductively the value vectors $\boldsymbol{z}^i, \hat{\boldsymbol{z}}^i$ at layer $1 \leq i \leq \ell$ with

$$\boldsymbol{z}^i = \boldsymbol{W}^i \cdot \hat{\boldsymbol{z}}^{i-1} + \boldsymbol{b}^i \qquad \hat{\boldsymbol{z}}^i = ReLU(\boldsymbol{z}^i).$$

The vector $\hat{\boldsymbol{z}}$ is called post-activation values, $\boldsymbol{z}$ is called pre-activation values, and $\boldsymbol{z}_j^i$ is used to call the $j$-th neuron in the $i$-th layer. For $\boldsymbol{x} = \boldsymbol{z}^0$ the (vector of) input, we denote by $f(\boldsymbol{x}) = \boldsymbol{z}^\ell$ the output. Finally, pre- and post-activation neurons are called *nodes*, and when we refer to a specific node/neuron, we use $a, b, c, d, n$ to denote them, and $W_{a,b} \in \mathbb{R}$ to denote the weight from neuron $a$ to $b$. Similarly, for input $\boldsymbol{x}$, we denote by $\text{value}_{\boldsymbol{x}}(a)$ the value of neuron $a$ when the input is $\boldsymbol{x}$. A path $\pi$ is a sequence $\pi = (a_i)_{k \leq i \leq k'}$ of neurons in consecutive layers, and the weight of $\pi$ is $weight(\pi) = W_{a_k, a_{k+1}} \times \cdots \times W_{a_{k'-1}, a_{k'}}$.

Concerning the verification problem, we focus on the well studied local-robustness question. Local robustness asks to determine whether the output of a neural network will be affected under small perturbations to the input. Formally, for an input $\boldsymbol{x}$ perturbed by $\varepsilon > 0$ under distance $d$, then the DNN is locally $\varepsilon$-robust in $\boldsymbol{x}$ whenever:

$$\forall \boldsymbol{x'} \text{ s.t. } d(\boldsymbol{x}, \boldsymbol{x'}) \leq \varepsilon, \text{ we have } argmax_i(f(\boldsymbol{x'})[i]) = argmax_i(f(\boldsymbol{x})[i])$$

## 3 VALUE ABSTRACTION FOR DNN VERIFICATION

In this section, we describe different value (over-)abstractions on $\boldsymbol{z}$ that are used by efficient algorithms to certify robustness around an input $\boldsymbol{x}$. Over-abstractions of values include all values for $\boldsymbol{z}$ in the neighbourhood of $\boldsymbol{x}$, and thus a certificate for safety in the over-abstraction is a proof of safety for the original input $\boldsymbol{x}$.

### 3.1 THE BOX ABSTRACTIONS

The concept of value abstraction involves calculating upper and lower bounds for the values of certain neurons in a Deep Neural Network (DNN) when inputs fall within a specified range. This approach aims to assess the network's robustness without precisely computing the values for every input within that range.

Firstly, it's important to note that weighted sums represent a linear function, which can be explicitly expressed with relative ease. However, the ReLU (Rectified Linear Unit) function presents a challenge in terms of accurate representation. Although ReLU is a relatively straightforward piecewise linear function with two modes (one for $x < 0$ and another for $x \geq 0$), it is not linear. The complexity arises when considering the compounded effects of the ReLU function across the various layers of a ReLU DNN. It's worth noting that representing $\text{ReLU}(x)$ precisely is feasible when $x$ is "*stable*", meaning it's consistently positive or consistently negative, as there's only one linear

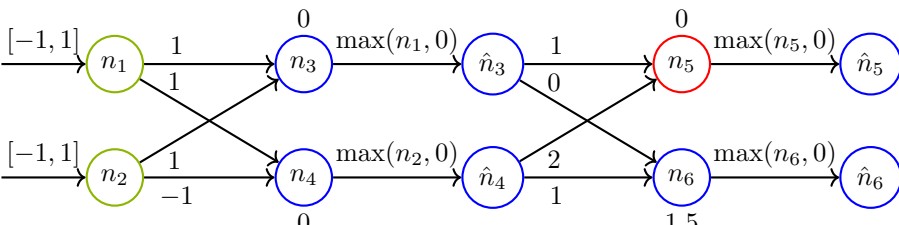

Figure 1: A DNN. Every neuron is separated into 2 nodes, $n$ pre- and $\hat{n}$ post-ReLU activation.

mode involved in each scenario. Consequently, the primary challenge lies in addressing "*unstable*" neurons, where the linearity of the function does not hold consistently.

Consider the simpler abstraction, termed "Box abstraction", recalled e.g. in Singh et al. (2019b): it inductively computes the bounds for each neuron in the subsequent layer independently. This is achieved by considering the weighted sum of the bounds from the previous layer, followed by clipping the lower bound at $\max(0, \text{lower bound})$ to represent the ReLU function, and so forth. For all $i \geq 3$, define $x_i = \text{value}_{\boldsymbol{x}}(n_i)$, where $\boldsymbol{x} = (x_1, x_2)$. Taking the DNN example from Fig 1, assume $x_1, x_2 \in [-1, 1]$. This implies that $x_3, x_4 \in [-2, 2]$. After applying the ReLU function, $\hat{x}_3, \hat{x}_4$ are constrained to $[0, 2]$, leading to $x_5 \in [0, 6]$ and $x_6 \in [0, 2]$. The bounds for $n_1, \ldots, n_4$ are exact, meaning for every $\alpha$ within the range, an input $\boldsymbol{y}$ can be found such that $\text{value}_{\boldsymbol{y}}(n_i) = \alpha$. However, this precision is lost from the next layer (beginning with $n_5, n_6$) due to potential dependencies among preceding neurons. For example, it is impossible for $x_5 = \text{value}_{\boldsymbol{x}}(n_5)$ to reach 6, as it would necessitate both $x_3 = 2$ and $x_4 = 2$, which is not possible at the same time as $x_3 = 2$ implies $x_1 = x_2 = 1$ and $x_4 = 2$ implies $x_2 = -1$ (and $x_1 = 1$), a contradiction.

In Ehlers (2017a); Singh et al. (2019b) and others, the *triangular abstraction* was proposed:

$$\text{ReLU}(x) = max(0, x) \leq \hat{x} \leq \text{UB}(n) \frac{x - \text{LB}(n)}{\text{UB}(n) - \text{LB}(n)} \tag{1}$$

It has two lower bounds (the 0 and identity functions), and one upper bound. DeepPoly Singh et al. (2019b) chooses one of the two lower bounds for each neuron $x$, giving rise to a greedy quadratic-time algorithm to compute very fast an abstraction of the value of $\hat{x}$ (but not that accurately).

### 3.2 MILP, LP AND PARTIAL MILP ENCODINGS FOR DNNs

At the other end of the spectrum, we find the Mixed Integer Linear Programming (MILP) value abstraction, which is a complete (but inefficient) method. Consider an unstable neuron $n$, whose value $x \in [\text{LB}(n), \text{UB}(n)]$ with $\text{LB}(n) < 0 < \text{UB}(n)$. The value $\hat{x}$ of $\text{ReLU}(x)$ can be encoded exactly in an MILP formula with one integer (actually even binary) variable $a$ valued in $\{0, 1\}$, using constants $\text{UB}(n), \text{LB}(n)$ with 4 constraints Tjeng et al. (2019):

$$\hat{x} \geq x \quad \wedge \quad \hat{x} \geq 0, \quad \wedge \quad \hat{x} \leq \text{LB}(n) \cdot a \quad \wedge \quad \hat{x} \leq x - \text{UB}(n) \cdot (1 - a) \tag{2}$$

For all $x \in [\text{LB}(n), \text{UB}(n)] \setminus 0$, there exists a unique solution $(a, \hat{x})$ that meets these constraints, with $\hat{x} = \text{ReLU}(x)$ Tjeng et al. (2019). The value of $a$ is 0 if $x < 0$, and 1 if $x > 0$, and can be either if $x = 0$. This encoding approach can be applied to every (unstable) ReLU node, and optimizing its value can help getting more accurate bounds. However, for networks with hundreds of *unstable* nodes, the resulting MILP formulation will contain numerous integer variables and generally bounds obtained will not be accurate, even using powerful commercial solvers such as Gurobi.

MILP instances can be linearly relaxed into LP over-abstraction, where variables originally restricted to integers in $\{0, 1\}$ (binary) are relaxed to real numbers in the interval $[0, 1]$, while maintaining the same encoding. As solving LP instances is polynomial time, this optimization is significantly more efficient. However, this efficiency comes at the cost of precision, often resulting in less stringent bounds. This approach is termed the *LP abstraction*.

In this paper, we propose to use *partial MILP*, to get interesting trade-offs between accuracy and runtime: for a set of unstable neurons $X$, we denote by $\text{MILP}_X$ the MILP encoding where variables

encoding $X$ are binary, and other variables are linear variables using the LP relaxation. We say that nodes in $X$ are *opened*. To further limit the number of binary variables needed for a given accuracy, we devise the same iterative approach as the box abstraction or DeepPoly Singh et al. (2019b), computing lower and upper bounds $\text{LB}(n), \text{UB}(n)$ for neurons $n$ of a layer, that are used when computing values of the next layer, thus necessitating less variables from previous layers.

The crucial factor in such an approach is to *select* few opened ReLU nodes in $X$ which are the most important for the accuracy. An extreme strategy was adopted in Huang et al. (2020), where only ReLU nodes of the immediate previous layer can be opened, and the measure to choose ReLU $a$ when computing the bounds for neuron $b$ was to consider $|W_{ab}|(\text{UB}(a) - \text{LB}(a))$. To obtain a more accurate measure, that is not limited to open nodes from the immediate previous layer, we invoke a folklore result on the LP relaxation of (2), for which we provide a direct and explicit proof:

**Proposition 1.** *The LP relaxation of (2) is equivalent with the triangular abstraction (1).*

*Proof.* Consider an unstable neuron $n$, that is $\text{LB}(n) < 0 < \text{UB}(n)$. The lower bound on $\hat{x}$ is simple, as $\hat{x} \geq 0 \wedge \hat{x} \geq x$ is immediatly equivalent with $\hat{x} \geq \text{ReLU}(x)$.

We now show that the three constraints $\hat{x} \leq \text{UB}(n) \cdot a \wedge \hat{x} \leq x - \text{LB}(n) \cdot (1-a) \wedge a \in [0,1]$ translates into $\hat{x} \leq \text{UB}(n) \frac{x - \text{LB}(n)}{\text{UB}(n) - \text{LB}(n)}$. We have $\hat{x}$ is upper bounded by $max_{a \in [0,1]}(min(\text{UB}(n) \cdot a, x - \text{LB}(n)(1-a)))$, and this bound can be reached. Furthermore, using standard function analysis tools (derivative...), we can show that the function $a \mapsto min(\text{UB}(n) \cdot a, x - \text{LB}(n)(1-a))$ attains its maximum when $\text{UB}(n) \cdot a = x - \text{LB}(n)(1-a)$, leading to the equation $(\text{UB}(n) - \text{LB}(n))a = x - \text{LB}(n)$ and consequently $a = \frac{x - \text{LB}(n)}{\text{UB}(n) - \text{LB}(n)}$. This results in an upper bound $\hat{x} \leq \text{UB}(n) \frac{x - \text{LB}(n)}{\text{UB}(n) - \text{LB}(n)}$, which can be reached, hence the equivalence. $\square$

## 4 UTILITY FUNCTION CHOOSING NEURONS IMPORTANT FOR ACCURACY.

In this section, we evaluate how each ReLU would impact the accuracy if encoded as a binary or a linear variable, using Proposition 1. A ReLU is said *open* when it is represented as a binary variable. For $X$ a set of open ReLUs, we denote by $\mathcal{M}_X$ the MILP model where variables from $X$ are encoded with binary variables, and other variables are using the LP linear relaxation.

Usually, heuristics to choose $X$ are based on evaluating the *sensitivity* of a neuron $z$ wrt the ReLU nodes, that is how much a ReLU value impacts the value of $z$, and rank the ReLU nodes accordingly. This is the case of Huang et al. (2020), but also more generally of heuristics for BaB, such as SR Bunel et al. (2020) and FSB De Palma et al. (2021). Instead, Utility considers the *improvement* from opening a neuron $n$, that is the difference for the value of $z$ between considering $\text{ReLU}(n)$ exactly or using its LP relaxation $\text{LP}(n)$. Indeed, it is not rare that $z$ is sensitive to ReLU node $n$, and yet $\text{LP}(n)$ already provides an accurate approximation of $\text{ReLU}(n)$. In this case, usual heuristics would open $n$, while it would only improve the value of $z$ in a limited way.

Assume that we want to compute an upper bound for neuron $z$ on layer $\ell_z$. We write $n < z$ if neuron $n$ is on a layer before $\ell_z$, and $n \leq z$ if $n < z$ or $n = z$. We denote $(\text{Sol\_max}_X^z(n))_{n \leq z}$ a solution of $\mathcal{M}_X$ maximizing $z$. In particular, $\text{Sol\_max}_X^z(z)$ is the maximum of $z$ under $\mathcal{M}_X$.

Consider $(sol(n))_{n \leq z} = (\text{Sol\_max}_\emptyset^z(n))_{n \leq z}$, a solution maximizing the value for $z$ when all ReLU use the LP relaxation. Function $\text{Improve\_max}^z(n) = sol(z) - \text{Sol\_max}_{\{n\}}^z(z)$, accurately represents how much opening neuron $n < z$ reduces the maximum computed for $z$ compared with using only LP. We have $\text{Improve\_max}^z(n) \geq 0$ as $\text{Sol\_max}_{\{n\}}^z$ fulfills all the constraints of $\mathcal{M}_\emptyset$, so $\text{Sol\_max}_{\{n\}}^z(z) \leq sol(z)$. Similarly, we define $(\text{Sol\_min}_\emptyset^z(n))_{n \leq z}$ and $\text{Improve\_min}^z(n)$. Calling MILP on $\mathcal{M}_{\{n\}}$ for every neuron $n \leq z$ would however be very time consuming when the number of neurons $a$ to evaluate is large. The main novelty of our Utility function is that it uses a (single) LP call to compute $(sol(n))_{n \leq z}$, with negligible runtime wrt the forthcoming $\text{MILP}_X$ call, and yet accurately approximates $\text{Improve\_max}^z(n)$ to choose a meaningful set $X$ of open nodes (Table 6).

For a neuron $b$ on the layer before the layer $\ell_z$, we define:

$$\text{Utility\_max}^z(b) = W_{bz} \times (sol(\hat{b}) - \text{ReLU}(sol(b)))$$

Consider $b$ with $W_{bz} < 0$: to maximize $z$, the value of $\text{sol}(\hat{b})$ is minimized, which is $\text{sol}(\hat{b}) = \text{ReLU}(\text{sol}(b))$ thanks to Proposition 1. Even if $z$ is sensitive to this ReLU $b$, the improvement of $b$ is 0. Utility does not open it as $\text{Utility\_max}^z(b) = 0$, whereas usual heuristics would.

For a neuron $a$ $a$ two layers before $\ell_z$, $b$ denoting neurons in the layer $\ell$ just before $\ell_z$, we define:

$$\Delta(\hat{a}) = \text{ReLU}(\text{sol}(a)) - \text{sol}(\hat{a})$$
$$\forall b \in \ell, \Delta(b) = W_{ab}\Delta(\hat{a})$$
$$\forall b \in \ell, \Delta(\hat{b}) = \begin{cases} \frac{\text{UB}(b)}{\text{UB}(b) - \text{LB}(b)}\Delta(b), & \text{if } W_{bz} > 0 \\ \max(\Delta(b), -\text{sol}(b)), & \text{if } W_{bz} < 0 \text{ and } \text{sol}(b) \geq 0 \\ \max(\Delta(b) + \text{sol}(b), 0), & \text{if } W_{bz} < 0 \text{ and } \text{sol}(b) < 0 \end{cases}$$
$$\text{Utility\_max}^z(a) = -\sum_{b \in \ell} W_{bz}\Delta(\hat{b})$$

Informally, $\Delta(\hat{a}), \Delta(b), \Delta(\hat{b})$ approximate the improvement on the accuracy of $\hat{a}, b, \hat{b}$ when computing $\text{ReLU}(a)$ using the exact MILP encoding instead of LP. Using Proposition 1, we show:

**Proposition 2.** $0 \leq \text{Improve\_max}^z(a) \leq \text{Utility\_max}^z(a)$.

Thus, $\text{Utility\_max}^z(a)$ can be used to approximate $\text{Improve\_max}^z(a)$. In particular, for all nodes $a$ with $W_{az} < 0$, this node will have the smallest $\text{Utility\_max}^z(a) = 0$ (thus will not get picked in the open nodes $X$), and indeed it is not having any impact on $\text{Sol\_max}^z_{\{a\}}(z)$. This is one striking difference (but not the only one) with choosing utility based on $|W_{az}|$ Huang et al. (2020).

*Proof.* Consider $\text{sol}'(n)_{n \leq z}$ with $\text{sol}'(n) = \text{sol}(n)$ for all $n \notin \{z, \hat{a}\} \cup \{b, \hat{b} \mid b \in \ell\}$. In particular, $\text{sol}'(a) = \text{sol}(a)$. Now, define $\text{sol}'(\hat{a}) = \text{ReLU}(\text{sol}(a))$. That is, $\text{sol}'(\hat{a})$ is the correct value for $\hat{a}$, obtained if we open neuron $a$, compared to the LP abstraction for $\text{sol}(\hat{a})$. We define $\text{sol}'(b) = \text{sol}(b) + \Delta(b)$ and $\text{sol}'(\hat{b}) = \text{sol}(\hat{b}) + \Delta(\hat{b})$. Last, $\text{sol}'(z) = \text{sol}(z) + \sum_{b \in \ell} W_{bz}\Delta(\hat{b})$. We will show:

$$(\text{sol}'(n))_{n \leq z} \text{ satisfies the constraints in } \mathcal{M}_{\{a\}} \tag{3}$$

This suffices to conclude: as $\text{sol}'(z)$ is a solution of $\mathcal{M}_{\{a\}}$, it is smaller or equal to the maximal solution: $\text{sol}'(z) \leq \text{Sol\_max}^z_{\{a\}}(z)$. That is, $\text{sol}(z) - \text{sol}'(z) \geq \text{sol}(z) - \text{Sol\_max}^z_{\{a\}}(z)$, i.e. $\text{Utility\_max}^z(a) \geq \text{Improve\_max}^z(a)$. In particular, we have that $\text{Utility\_max}^z(a) \geq 0$, which was not obvious from the definition.

Finally, we show (3). First, opening $a$ changes the value of $\hat{a}$ from $\text{sol}(\hat{a})$ to $\text{ReLU}(\text{sol}(a)) = \text{sol}(\hat{a}) + \Delta(a)$, and from $\text{sol}(b)$ to $\text{sol}(b) + \Delta(b)$. The case of $\Delta(\hat{b})$ is the most interesting: If $W_{bz} > 0$, then according to Proposition 1, the LP solver sets $\text{sol}(\hat{b}) = \text{sol}(b)\frac{\text{UB}(b)}{\text{UB}(b) - \text{LB}(b)} + \text{Cst}$ to maximize $z$. Changing $b$ by $\Delta(b)$ thus results in changing $\text{sol}(\hat{b})$ by $\frac{\text{UB}(b)}{\text{UB}(b) - \text{LB}(b)}\Delta(b)$. If $W_{bz} \leq 0$, then the LP solver sets $\text{sol}(\hat{b})$ to the lowest possible value to maximize $z$, which happens to be $\text{ReLU}(b)$ according to Proposition 1. If $\text{sol}(b) < 0$, then we have $\text{sol}(\hat{b}) = \text{ReLU}(b) = 0$ and opening $a$ change the 0 value only if $\text{sol}(b) + \Delta(b) > 0$. If $\text{sol}(b) > 0$, then $\text{sol}(\hat{b}) = \text{ReLU}(\text{sol}(b)) = \text{sol}(b)$, and the change to $\hat{b}$ will be the full $\Delta(b)$, unless $\Delta(b) < -\text{sol}(b) < 0$ in which case it is $-\text{sol}(b)$. $\square$

We can proceed inductively in the same way to define $\text{Utility\_max}^z(a)$ for deeper neurons $a$.

## 5 EXPERIMENTAL EVALUATION

We implemented Hybrid MILP in Python 3.8, and Gurobi 9.52 was used for solving LP and MILP problems. We conducted our evaluation on an AMD Threadripper 7970X (32 cores@4.0GHz, 5nm) with 256 GB of main memory and 2 NVIDIA RTX 4090.

The objectives of our evaluation was to answer the following questions:

1. How does the the choice of the set $X$ impacts the accuracy of MILP$_X$?

2. How accurate is Hybrid MILP, and how efficient is it?

## 5.1 EVALUATION OF THE UTILITY FUNCTION TO CHOOSE NEURONS TO OPEN

To measure the impact of the utility function to select neurons to open, we focused on a small hard DNN, namely $5 \times 100$, so as to be able to compute exact bounds for the first few layers using a full MILP encoding of the DNN for comparison purpose. We tested over the $x = 59$th image in the MNIST dataset, as it has a large number of unstable ReLU nodes in the first few layers (61 in the first and 55 in the second layer), so we can experiment with a larger choice of values) for $K = |X|$ the size of set $X$. To measure the accuracy, we measure the uncertainty of all nodes in a layer: the uncertainty of a node is the range between its computed lower and upper bound. We then average the uncertainty among all the nodes of the layer. Formally, the uncertainty of a node $a$ with bounds $[\text{LB}, \text{UB}]$ is uncert$(a) = \text{UB} - \text{LB}$. The average uncertainty of layer $\ell$ is $\dfrac{\sum_{a \in l} \text{uncert}(a)}{size(\ell)}$.

We focus on the uncertainty of nodes in the third layers, wrt ReLU nodes in the first and second layer. The bounds for nodes of the first two layers are computed exactly using the full MILP encoding. We report in Figure 2 the average uncertainty of MILP$_X$ following the choice of the $K$ heaviest neurons for our **Utility** function, compared with a random choice, both for nodes exclusively from the previous Layer 2 or from both Layers 1 and 2. We compared with choosing based on strength$(n) = (\text{UB}(n) - \text{LB}(n)) \cdot |W_{nz}|$ Huang et al. (2020), opening nodes in Layer 2 only.

The Utility function selects very important neurons: to achieve the same accuracy than Huang et al. (2020), 2.5 time fewer nodes (10 vs 25) are necessary when picking in the same previous Layer 2. The ability from **Utility** to compare neurons from different layers enables even better frugality: 4 time fewer nodes (5 vs 20, 10 vs 40) are necessary to reach the same accuracy than Huang et al. (2020). Overall, choosing 35 neurons by **Utility** improves accuracy by $95\%$ of what can be done if all $|X| = 116$ nodes are opened compared with LP (corresponding to $|X| = 0$).

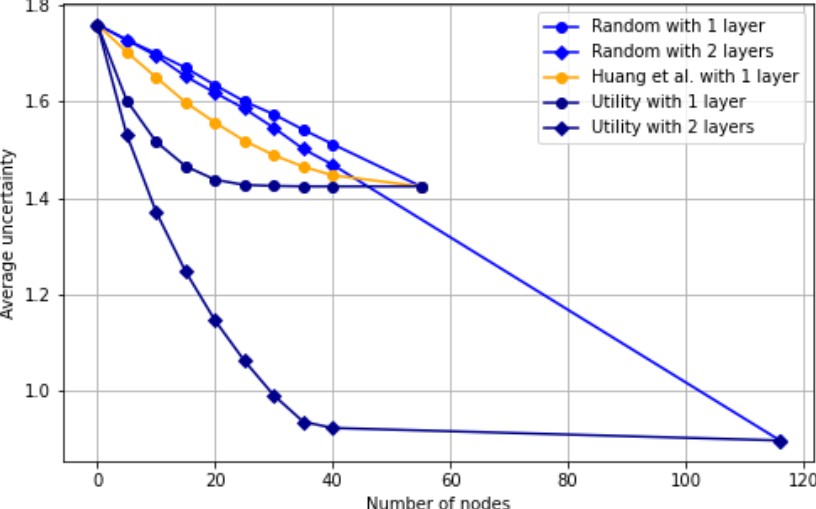

Figure 2: Average uncertainty of MILP$_X$ for nodes of the third layer, for $X$ with $K$ ReLU nodes of the (1st and) 2nd layer, chosen by our **Utility** function vs Huang et al. (2020) vs random choice.

## 5.2 COMPARISON WITH $\alpha, \beta$-CROWN

We conducted our evaluation on the neural networks tested in Wang et al. (2021) which display a large number ($\geq 20\%$) of images undecided by $\alpha, \beta$-Crown. That is, these DNNs are *hard* to verify. Namely, these are 6 ReLU-DNNs: 5 MNIST DNN that can be found in the ERAN GitHub (the 4th to the 8th DNNs provided) as well as 1 CIFAR CNN from Balunovic & Vechev (2020), see

| Network | $\alpha, \beta$-Crown TO=10s | $\alpha, \beta$-Crown TO=30s | $\alpha, \beta$-Crown TO=2000s | Refined $\beta$-Crown | **Hybrid MILP** |
|---|---|---|---|---|---|
| MNIST 5×100 | 57% (6.9s) | 55% (18.9s) | 50% (1026s) | 13% (92s) | 13% (**46s**) |
| MNIST 5×200 | 50% (6.5s) | 47% (17s) | 46% (930s) | 9% (80s) | **8% (71s)** |
| MNIST 8×100 | 63% (7.2s) | 58% (20s) | 58% (1163s) | 21% (102s) | **15% (61s)** |
| MNIST 8×200 | 56% (6.8s) | 55% (18s) | 54% (1083s) | 16% (83s) | **8% (78s)** |
| MNIST 6×500 | 53% (6.4s) | 51% (16s) | 50% (1002s) | – | **10% (402s)** |
| CIFAR CNN-B-adv | 28% (4.3s) | 22% (8.7s) | 20% (373s) | – | **11% (417s)** |
| CIFAR ResNet | 0% (2s) | 0% (2s) | 0% (2s) | – | 0% (2s) |

Table 3: Undecided images (%, *lower is better*) as computed by $\alpha, \beta$-Crown, Refined $\beta$-Crown and Hybrid MILP on 7 DNNs (average runtime per image). The 6 first DNNs are hard instances. The last DNN (ResNet) is an easy instance (trained using Wong to be easy to verify, but with a very low accuracy level), provided for reference.

also Dathathri et al. (2020), which can be downloaded from the $\alpha, \beta$-Crown GitHub. We commit to the same $\epsilon$ settings as in Wang et al. (2021), that are recalled in Table 1. For reference, we also report an easy but very large ResNet Network for CIFAR10, already tested with $\alpha, \beta$ CROWN. We report in Table 3 the % of undecided images, that is the % of images than can be neither falsified (by $\alpha, \beta$-CROWN) nor verified by the tested verifier, among the 100 first images for each MNIST or CIFAR10 benchmark. The exact same DNNs and $\epsilon$ are used in Tables 1 and 3.

*Analysis*: on easy instances, Hybrid MILP is virtually similar to $\alpha, \beta$-CROWN, as it is called first and it is sufficient to have 0% undecided images, as shown even on the very large ResNet.

On hard instances (the 6 first DNNs tested), Hybrid MILP is very accurate, only leaving 8%-15% of images undecided, with runtime taking less than $500s$ in average per image, and even 10 times less on smaller DNNs. It can scale up to quite large hard DNNs, such as CNN-B-Adv with 2M parameters and 20K activations.

Compared with $\alpha, \beta$-Crown with a time-out of TO=2000s, it is much more accurate, with a reduction of undecided images by $9\% - 43\%$. It is also from 20x faster on smaller networks to similar time on the largest DNN. Compared with $\alpha, \beta$-Crown with a time-out of TO=30s, the accuracy gap is even larger (e.g. 11% for CNN-B-Adv, i.e. half undecided images), although the average runtime is also obviously larger (solving hard istances takes longer than solving easy instances).

Last, compared with *Refined* $\beta$-Crown, we can observe three patterns: on the shallowest DNNs (5×100, 5×200), Refined $\beta$-Crown can run full MILP on almost all nodes, reaching almost the same accuracy than Hybrid MILP, but with longer runtime (up to 2 times on 5×100). On intermediate DNNs (8×100, 8×200), full MILP invoked by Refined $\beta$-Crown can only be run on a fraction of the neurons, and the accuracy is not as good as Hybrid MILP, with $6\% - 8\%$ more undecided images (that is double on 8×200), while having longer runtime. Last but not least, Refined $\beta$-Crown cannot scale to larger instances (6×500, CNN-B-Adv), while Hybrid MILP can.

### 5.3 FINER GRAIN EVALUATION OF ACCURACY

In order to evaluate the accuracy of Hybrid MILP in a finer way, showcasing the capabilities to have a very accurate and efficient verifier, we consider a quantitative questions for each image, rather than a pure yes (verified)/ no (not verified) question. Namely, we compute the $\epsilon$ for which the verifier can certify local-robustness around a given image, which makes sense as there is little rationale in setting a particular $\epsilon$ (which is however the usual local-robustness setting).

For that, we considered the challenging DNN MNIST $6 \times 500$, and the 20 first images from the MNIST benchmark. We first run the attack from $\alpha, \beta$-Crown for varying $\epsilon$, using a binary search, to set up an upper bound on $\epsilon$ (the initialization is $\epsilon \in [0, 1]$). Then we run binary search with a global time-out of $10000s$, initialized from 0 to this upper bound, where each call is either to $\alpha, \beta$-Crown with TO=2000s, or Hybrid MILP. We report the results (upper bound, best bound verified by $\alpha, \beta$-Crown and by Hybrid MILP) for each image. We report the results in Fig. 3.

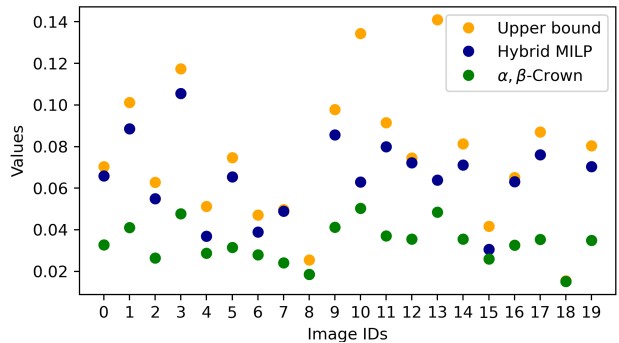

.

Figure 3: $\epsilon$-robustness proved after $10000s$ for $6 \times 500$ on each of the 20 first images of MNIST.

*Analysis*: in 90% of the cases, Hybrid MILP is very close to the upper bound. On 2 images (10 and 13), Hybrid MILP is far from the upper bound: these are also the ones where the upper bound is the highest, so it is possible that the falsifier missed a closer attack to robustness.

Compared with $\alpha, \beta$-CROWN, Hybrid MILP is much closer to the upper bound, except for 2 cases (images 8 and 18) where $\alpha, \beta$-CROWN is already very close to the upper bound. On average, the $\epsilon$-gap to upper bound is $0.014$ for Hybrid MILP, 3 times smaller than the $0.042$ gap for $\alpha, \beta$-CROWN.

### 5.4 COMPARISON WITH OTHER VERIFIERS?

We voluntarily limited the comparison to $\alpha, \beta$-Crown because it is the most efficient verifier to date, and to consider a spectre of parameters to understand $\alpha, \beta$-Crown scaling without too much clutter.

Notice that results for other verifiers (PRIMA Müller et al. (2022), SDP-FO Dathathri et al. (2020), etc) were already reported on these DNNs Wang et al. (2021), with unfavorable comparison vs $\alpha, \beta$-CROWN. Further, we reported accuracy of complete verifiers, NNenum Bak (2021), Marabou Katz et al. (2019); Wu et al. (2024), respectively 4th, 3rd of the last VNNcomp'24 Brix et al. (2024), as well as full MILP Tjeng et al. (2019) in Table 2, showing that these verifiers are not competitive on hard instances either, even on the smallest hard DNN. Concerning MnBAB Ferrari et al. (2022), we tested it in appendix, and it compares slightly unfavorably in time and accuracy towards $\alpha, \beta$-CROWN on CNN-B-Adv and *hard* MNIST DNNs at several time-out settings. Last, Pyrat Durand et al. (2022) (2sd in the latest VNNComp) is not open source, which made running it impossible.

## 6 CONCLUSION

In this paper, we developed a novel Utility function to select few ReLU nodes to consider with binary variables to compute accurately bounds on neurons of DNNs. The novelties are that it focuses on *improvement* wrt a given input, rather than on generic *sensitivity* of a neuron wrt to a ReLU node, and it uses the solution of one call to an (efficient LP) solver to evaluate this improvement. This makes the choice particularly efficient, necessitating $\approx 4x$ less integer variables than previous proposals Huang et al. (2020) for the same accuracy. Our empirical studies revealed that this can yield highly accurate results, verifying up to 40% more images than the SOTA ($\alpha, \beta$-Crown, winner of the 4 last VNNComp), with the same runtime, for DNNs with up to 20 000 neurons. The reason is that $\alpha, \beta$-Crown hits a complexity barrier, similarly as other competing solutions, when considering hard (even small) DNNs. This opens a lot of perspectives, among which: verifying efficiently other hard instances; certifying $\epsilon$-robustness of images for $\epsilon$ as large as possible; verifying global rather than local properties Wang et al. (2022).

**Reproducibility Statement:** We tested twice outlier results to confirm them, making sure of reproducibility on the given hardware. Precise details on the settings used are provided in the appendix. Additional results (e.g. ablation studies) are also provided in the appendix. Tested DNNs as well as MNIST and CIFAR10 DataSet are freely available. The source code of Hybrid MILP will be provided on GitHub after acceptance (needing Gurobi as well as $\alpha, \beta$-Crown).

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

APPENDIX

## A  PARAMETER SETTINGS

### SETTING FOR $\alpha, \beta$-CROWN

The networks were already tested by $\alpha, \beta$-Crown Wang et al. (2021). We thus simply reused the parameter files from their Github, except for time-out which we explicitly mention.

e.g., for CNN-B-Adv: "solver: batch size: 512 beta-crown: iteration: 20" and for MNIST 5x100: "solver: batch size: 1024 beta-crown: iteration: 20".

We did not experiment with cutting planes (GCP-CROWN Zhang et al. (2022)), as it needs an additional package, namely IBM CPLEX solver, we do not have access to. From Zhang et al. (2022), the number of undecided inputs of GCP-CROWN is $\leq 2\%$ better than $\alpha, \beta$-Crown on the DNNs we experimented with, far from the $10 - 40\%$ improvement seen from Hybrid MILP. The conclusion are thus unchanged.

### SETTING FOR HYBRID MILP

Hybird MILP first call $\alpha, \beta$-Crown with short time-out (TO), then call partial MILP on those inputs which was neither certified nor falsified by this run of $\alpha, \beta$-Crown. We are using two settings of TO, for smaller DNNs we use T0= $10s$, and for the two larger ones, we use TO= $30s$.

The setting for partial MILP for fully-connected DNNs is about how many neurons need to be opened (once set, the selection is automatic). The runtime depending crucially upon the number of open ReLU neurons, we set it quite tightly, only allowing few neuron deviation to accommodate to a particularly accurate/inaccurate bound computation (measure by the weight of the remaining Utility function). As complexity increases with the layer considered, as the size of the MILP model grows, we lower this number with the depth, only committing to an intermediate number for the output neuron (the number of output neurons is smaller than hidden layer, and this is the most important computation). We experimentally set this number so that each computing the bounds in each hidden layer takes around the same time. Remember that in layer 1, partial MILP is not necessary and propagating bounds using interval arithmetic is already exact. We open [48,48] to compute bounds for hidden layer 2, [21,24] for layer 3, [11,14] for layer 4, [6,9] for layer 5, [3,6] for layer 6, [2,5] for layer 7, [1,4] for hidden layer 8 (if any), and we open [14,17] for the output layer. The exact number of open nodes in the range [a,a+3] is decided automatically for each neuron being computed : ReLUs are ranked according to their value by Utility, and the a top ReLUs are open. Then, ReLUs ranked a+1,a+2, a+3 are opened if their Utility value is larger than a small threshold. We set the threshold at 0.01. It should be seen as a way to save runtime when Utility knows that the next node by ranking (a+i) will not impact accuracy much (thanks to the upper bound from Proposition 2).

| Network | TO for $\alpha, \beta$-Crown | Minimum number of Open neurons |
|---|---|---|
| MNIST $5 \times 100$ | 10s | 48,21,11,6,14 |
| MNIST $5 \times 200$ | 10s | 48,21,11,6,14 |
| MNIST $8 \times 100$ | 10s | 48,21,11,6,3,2,1,14 |
| MNIST $8 \times 200$ | 10s | 48,21,11,6,3,2,1,14 |
| MNIST $6 \times 500$ | 30s | 48,21,11,6,3,14 |
| CIFAR CNN-B-adv | 30s | 200, 0, 45 |

Table 4: Settings of Hybrid MILP for the different *hard* instances

For convolutional CNNs, the strategy is adapted, as there is much more neurons, but in a shallower architecture and not fully connected. The second layer is computed accurately, opening 200 neurons, which is manageable as there is only one ReLU layer to consider, and accuracy here is crucial. We do not open any nodes in the third layer (the first fully connected layer) if the output layer is the next one (which is the case for CNN-B-Adv), and instead rely on the choice of important nodes for the output layer. Otherwise, we open 20 neurons. In the output layer, we open at least 45 neurons (there is less output neurons than nodes in the previous layer), and enlarge the number of open neurons (up to 300) till we find an upper bound, that is a best current MILP solution, of around +0.1 (this 0.1 was experimentally set as target, a good balance between accuracy and efficiency), and compute a guaranteed lower bound (the goal is to guarantee the bound is $> 0$).

In Table 4, we sum-up the TO and minimum open numbers for each DNN considered.

$\alpha, \beta$-Crown uses massively parallel ($>4096$ threads) GPU, while Partial MILP uses 20 CPU-threads.

Notice that a different balance between accuracy and runtime could be set. For instance, we set up the numbers of open neurons to have similar runtime as Refined $\beta$-Crown for the first 4 DNNs ($50s - 100s$). We could easily target better accuracy (e.g. for $8 \times 100$ with a relatively high $15\%$ undecided images) by increasing the number of open neurons, with a trade-off on runtime (current runtime is at $61s$). By comparison, the sweet spot for $\alpha, \beta$-Crown seems to be around TO$= 30s$, enlarging the time-out having very little impact on accuracy but large impact on runtime (Table 1).

Last, for Gurobi, we use a custom MIP-Gap (from 0.001 to 0.1) and time-out parameters, depending on the seen improvement and the possibility to make a node stable. This is low level implementation details that will be available in the code once the paper is accepted.

## B  PSEUDOCODE AND COMPLEXITY ANALYSIS

---

**Algorithm 1:** pMILP($K$)

---

**Input:** Bounds $[\alpha_n, \beta_n]$ for input nodes $n$ at layer 0 (input neighbourhood)
**Output:** Bounds $[\alpha_n, \beta_n]$ for every node $n$
1 **for** *layer $k = 1 \cdots \ell$* **do**
2     **for** *neuron $n$ in layer $k$* **do**
3        Compute $X$ a set of $K$ nodes with the highest Utility for target neuron $n$.
4        Run MILP$_X$ to obtain $[\alpha_n, \beta_n]$ from bounds of neurons in layers $< k$

---

We provide the pseudo code for pMILP in Algorithm 1. pMILP($K$) has a worst case complexity bounded by $O(N \cdot \text{MILP}(N, K))$, where $N$ is the number of nodes of the DNN, and MILP$(N, K)$ is the complexity of solving a MILP program with $K$ integer variables and $N$ linear variables. We have MILP$(N, K) \leq 2^K \text{LP}(N)$ where LP$(N)$ is the Polynomial time to solve a Linear Program with $N$ variables, $2^K$ being an upper bound. Solvers like Gurobi are quite adept and usually do not need to evaluate all $2^K$ ReLU configurations to deduce the bounds. It is worth mentioning that the "for" loop iterating over neurons $n$ in layer $k$ (line 2) can be executed in parallel, because the computation only depends on bounds from preceding layers, not the current layer $k$.

If $K$ is sufficiently small, this approach is expected to be efficient. The crucial part is thus to find few neurons which are particularly important when computing neuron $n$. This is where our novel Utility function outperforms previous solutions, by using an call to an (LP) solver to obtain a solution, which is novel as far as we know. This solutions allows to optimizing the choice of open nodes for the particular input considers, and appears significantly better than previous attempts, explaining the efficiency of the method.

For comparison, refined $\beta$-Crown and refined Prima used all the nodes up to a certain layer as binary variables, which is particularly inefficient, see Table 8 and Fig. 7. Hence, it can only be applied to small DNNs (it is implemented only up to MNNIST 8x200), while pMILP scales to at least 20.000 neurons (CNN-B-Adv). Huang et al. (2020) is the closest to pMILP, also implementing a choice of nodes as binary variables. However, their choice is particularly inefficient, as revealed by Table 6, needing 4 times the number of open nodes for the same accuracy as our Utility function.

## C Ablation studies

In this section, we consider ablation studies to understand how each feature enables the efficiency of pMILP.

### Time scaling with open nodes

First, we explore the time scaling with different number of open nodes, for our full Utility function using nodes in the last two layers (Layer 1 and 2), providing finer details than in Table 3, with the same setting, i.e. previous layer being computed with full MILP.

| $|X|$ | Time | Uncertainty |
|---|---|---|
| 0 | 2.6 | 1.760946128 |
| 1 | 7.3 | 1.702986873 |
| 2 | 11.1 | 1.65469034 |
| 3 | 16.3 | 1.612137282 |
| 4 | 15.5 | 1.571001109 |
| 5 | 15.7 | 1.531925404 |
| 6 | 15.8 | 1.49535638 |
| 7 | 16.4 | 1.46189314 |
| 8 | 15.8 | 1.4299535 |
| 9 | 17.2 | 1.4006364 |
| 10 | 22.5 | 1.3711203 |
| 11 | 27.2 | 1.3438245 |
| 12 | 21.6 | 1.3183356 |
| 13 | 28.7 | 1.2938690 |
| 14 | 29.6 | 1.2690507 |
| 15 | 24.5 | 1.2475106 |

| $|X|$ | Time | Uncertainty |
|---|---|---|
| 16 | 31.9 | 1.2243065 |
| 17 | 28.6 | 1.2031791 |
| 18 | 30.4 | 1.1839474 |
| 19 | 34.0 | 1.1644653 |
| 20 | 42.1 | 1.1456181 |
| 21 | 47.6 | 1.1261252 |
| 22 | 62.7 | 1.1089745 |
| 23 | 70.0 | 1.0931242 |
| 24 | 70.8 | 1.0773088 |
| 25 | 139.9 | 1.060928 |
| 26 | 154.2 | 1.045715 |
| 27 | 213.1 | 1.030605 |
| 28 | 211.3 | 1.016058 |
| 29 | 373.1 | 1.001374 |
| max=116 | 3300 | 0.895 |

Table 5: Time and uncertainty scaling of pMILP with number of nodes.

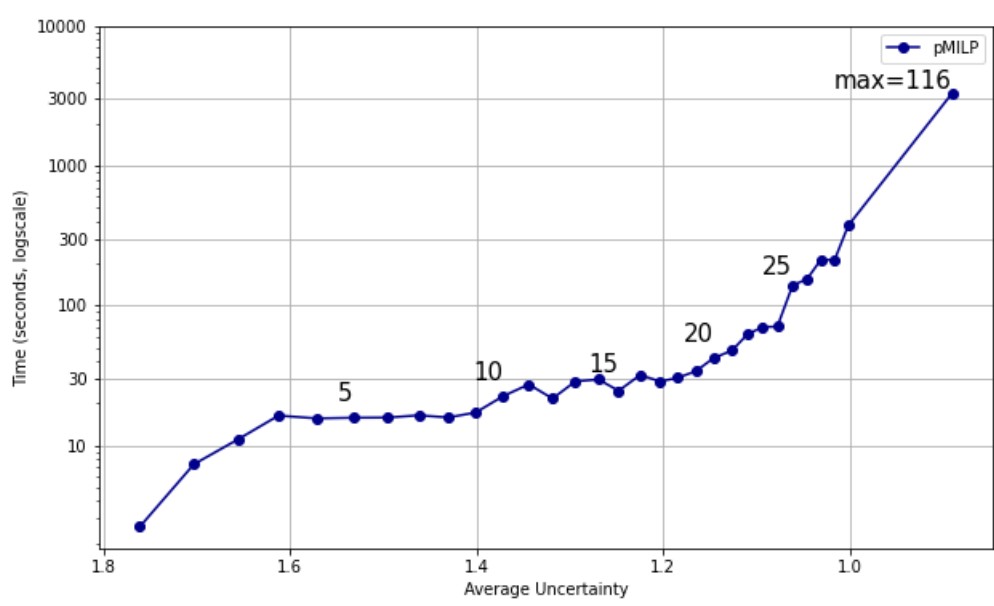

Figure 4: Time and uncertainty scaling of pMILP with number of nodes. Time is using logscale.

The exponential complexity with the number of nodes can be seen on Figure 4, where time is represented using logarithmic scale. The flat area in the middle is Gurobi having good heuristic to avoid considering all $2^K$ cases when $K < 21$ is not too large, but not working so well for $K > 25$. Notice that when certifying, pMILP uses $|X| \in$ 21-24, which is a good trade off between time and accuracy.

We also provide in Table 6 the raw numbers used to produce Figure 2. Further, we tested with the SR Bunel et al. (2020) and FSB heuristics De Palma et al. (2021), that chooses nodes to branch on for BaB (Branch and Bound). When SR and FSB are used to choose open nodes for pMILP, the accuracy is low as shown on Fig. 5: SR and FSB are worse than Huang et al. (2020) for $< 35$ open ReLU nodes, although unlike the latter, they can rank ReLU nodes in several layer before (which helps them a bit), and far worse than Utility. Further, FSB is performing worse than SR when choosing nodes for pMILP, while to choose nodes to branch on for BaB, it is the opposite De Palma et al. (2021). This likely means that the heuristic to choose nodes to branch for BaB is not adapted to choose nodes to open for pMILP.

| $|X|$ | $X \subseteq$ Layer 2, max $= 55$ | | | | | $X \subseteq$ Layers 1&2, max $= 116$ | | | |
| --- | --- | --- | --- | --- | --- | --- | --- | --- | --- |
| | Random | Huang | SR | FSB | **Utility** | Random | SR | FSB | **Utility** |
| 0 (LP) | 1.761 | 1.761 | 1.761 | 1.761 | 1.761 | 1.761 | 1.761 | 1.761 | 1.761 |
| 5 | 1.729 | 1.704 | 1.7200 | 1.7197 | 1.603 | 1.729 | 1.7133 | 1.7149 | **1.532** |
| 10 | 1.701 | 1.651 | 1.6840 | 1.6851 | 1.517 | 1.696 | 1.6674 | 1.6714 | **1.371** |
| 15 | 1.671 | 1.599 | 1.6502 | 1.6516 | 1.466 | 1.653 | 1.6230 | 1.6251 | **1.247** |
| 20 | 1.635 | 1.557 | 1.6190 | 1.6199 | 1.438 | 1.619 | 1.5764 | 1.5812 | **1.145** |
| 25 | 1.601 | 1.519 | 1.5887 | 1.5886 | 1.427 | 1.586 | 1.5322 | 1.5388 | **1.061** |
| 30 | 1.574 | 1.489 | 1.5584 | 1.5604 | 1.425 | 1.546 | 1.4914 | 1.4982 | **0.989** |
| 35 | 1.542 | 1.465 | 1.5289 | 1.5305 | 1.424 | 1.502 | 1.4481 | 1.4600 | **0.934** |
| 40 | 1.512 | 1.447 | 1.4985 | 1.5001 | 1.424 | 1.469 | 1.4070 | 1.4187 | **0.921** |
| max | 1.424 | 1.424 | 1.424 | 1.424 | 1.424 | 0.895 | 0.895 | 0.895 | 0.895 |

Table 6: Average uncertainty of MILP$_X$ for nodes of the third layer, with ReLU nodes of the (1st and) 2nd layer, chosen by our **Utility** function vs Huang et al. (2020) vs vs SR vs FSB vs random.

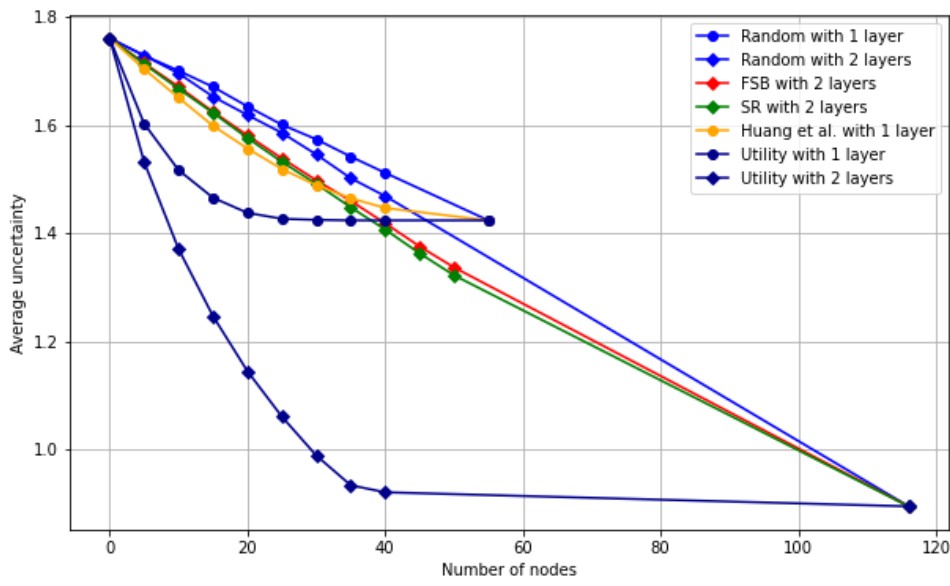

Figure 5: Average uncertainty of MILP$_X$ for nodes of the third layer, with ReLU nodes of the (1st and) 2nd layer, chosen by our **Utility** function vs Huang et al. (2020) vs vs SR vs FSB vs random.

Then, we explore the usefulness of computing accurately each layer inductively. For that, we keep the setting of Figure 4 / Table 5, but computing the previous layer with LP rather than with full MILP.

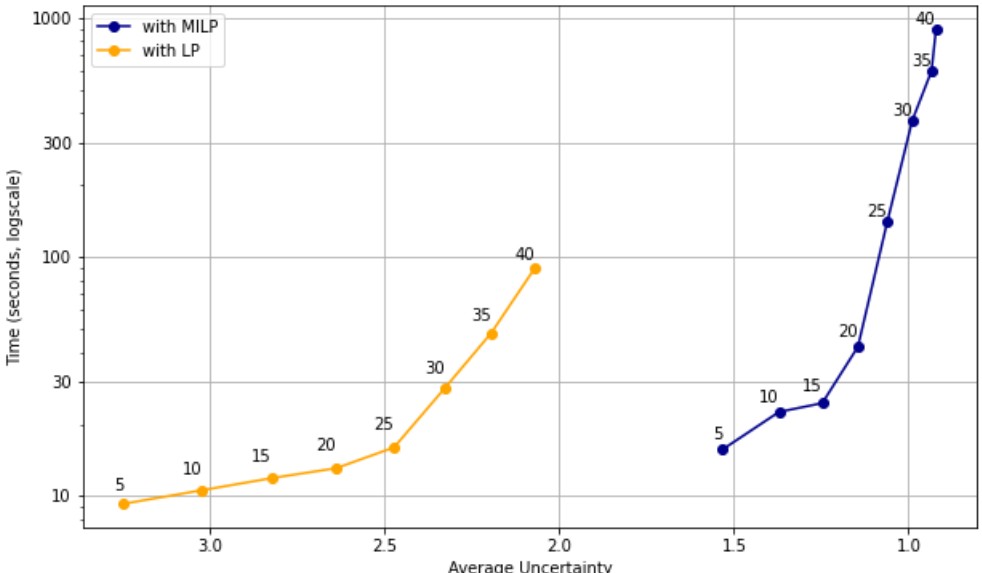

Figure 6: Comparison of accuracy in layer 3 when layer 2 is computed inaccurately using LP vs when layer 2 computed accurately using MILP. Time is using logscale.

| $|X|$ | Time | With LP for layer 2 | With MILP for layer 2 |
|-------|------|---------------------|-----------------------|
| 5 | 9.3 | 3.24737 | 1.532 |
| 10 | 10.6 | 3.02214 | 1.371 |
| 15 | 11.9 | 2.82383 | 1.247 |
| 20 | 13.1 | 2.63862 | 1.145 |
| 25 | 16.0 | 2.47324 | 1.061 |
| 30 | 28.3 | 2.32793 | 0.989 |
| 35 | 48.1 | 2.19506 | 0.934 |
| 40 | 89.4 | 2.07107 | 0.921 |

Table 7: Comparison of accuracy in layer 3 when layer 2 is computed inaccurately using LP vs when layer 2 computed accurately using MILP.

This experiment explains the rationale to use divide and conquer protocol, using many calls (one for each neuron) with relatively small number $|X|$ of open nodes rather than fewer calls to MILP with larger number $|X|$ of open nodes. This is clear already with only 1 layer before.

Running full MILP till a small MIP-Gap (typically 0.001) is reached is extremely time inefficient.

Instead, the standard strategy is to set a reasonable time-out and use whatever bound has been generated. We compare this standard strategy with the pMILP strategy of setting a priori a number of open nodes.

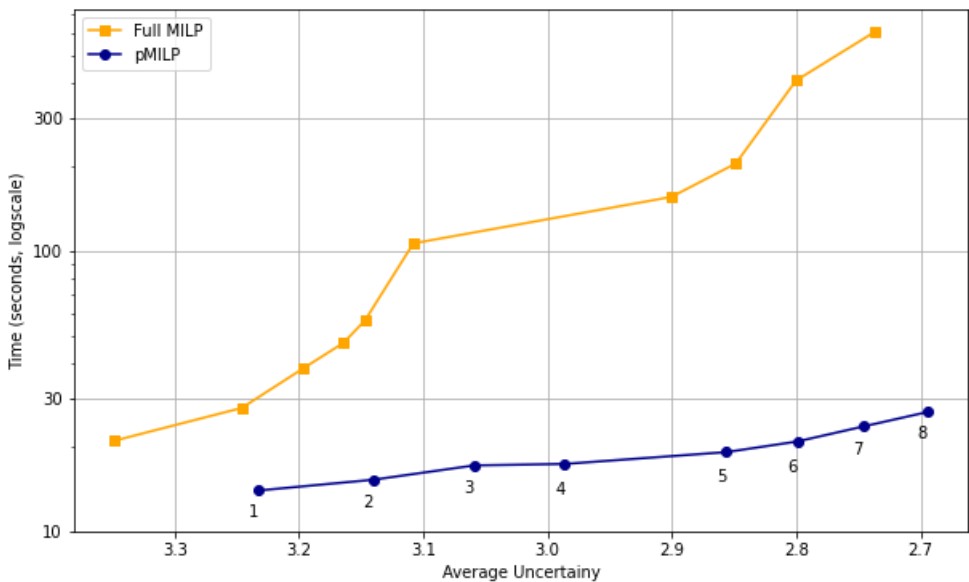

.

Figure 7: Comparison of uncertainty at layer 7 for full MILP with different time-outs vs pMILP with different number of open nodes. Time is using logscale.

(a) pMILP

| $|X|$ | Time | Uncertainty |
|---|---|---|
| 1 | 14 | 3.233021901 |
| 2 | 15.2 | 3.140309921 |
| 3 | 17.21 | 3.059083103 |
| 4 | 17.4 | 2.986166762 |
| 5 | 19.2 | 2.856229765 |
| 6 | 20.9 | 2.799248232 |
| 7 | 23.7 | 2.746167245 |
| 8 | 26.6 | 2.69485246 |

(b) full MILP

| Time | Uncertainty |
|---|---|
| 21.1 | 3.348236261 |
| 27.6 | 3.24604282 |
| 38.2 | 3.196640184 |
| 47.1 | 3.164298172 |
| 56.7 | 3.146913614 |
| 106.7 | 3.108035223 |
| 156.3 | 2.900438725 |
| 205.8 | 2.848648426 |
| 406.7 | 2.800268264 |
| 606.1 | 2.737064255 |

Table 8: Comparison of bounding the number of nodes for pMILP and using different time outs for full MILP. In both settings, lower and upper bounds of previous layers are the same (computed by pMILP).

pMILP obtains 2.8 accuracy in $< 21$ seconds (with 7 open nodes), while full MILP needs 400 seconds to obtain it, a 19x speed up. For 2.7 accuracy, the speedup is $>> 22$.

Figure 7 shows that choosing nodes is much more efficient for time/accuracy trade-off than setting time outs and use full MILP. And this is for the smallest DNN we considered (500 hidden neurons, far from the biggest 20k neuron DNN we experimented with)

# D COMPARISON WITH OTHER DNN VERIFIERS

In the following, we provide results comparing $\alpha, \beta$-Crown to other verifiers, to justify our use of $\alpha, \beta$-Crown as state of the art for efficient verifiers as main source of comparison to Hybrid MILP for hard DNN instance.

## COMPARISON $\alpha, \beta$-CROWN VS PRIMA

PRIMA Müller et al. (2022) is a major verifier in the ERAN toolkit. In Table 9, we report the comparison between PRIMA and $\alpha, \beta$-Crown, mainly from Wang et al. (2021). The setting is mainly similar from ours, but numbers are not perfectly comparable as the images tested are not exactly the same (1000 first or 200 first images for CNN-B-Adv), vs 100 first in Tables 3, 1. Also, time-out settings and hardware are slightly different. The overall picture is anyway the same.

Analysis: On the 4 smallest MNIST networks, PRIMA uses a refined path comparable with Refined $\beta$-Crown. However, it is slower and less accurate than Refined $\beta$-Crown. On larger *hard* networks, PRIMA has also more undecided images than $\alpha, \beta$-Crown, while the runtime is $> 5$ times larger. Hence, Hybrid MILP is more accurate than Prima with similar runtime or faster.

Notice that kPoly Singh et al. (2019a), OptC2V Tjandraatmadja et al. (2020), SDP-FO Dathathri et al. (2020) numbers were also reported in Wang et al. (2021) on these networks, with even more unfavorable results.

## COMPARISON $\alpha, \beta$-CROWN VS MN-BAB

MN-BaB Ferrari et al. (2022) is an improvement built over PRIMA, using a similar Branch and Bound technique as used in $\alpha, \beta$-Crown. Results in Ferrari et al. (2022) are close to those of $\alpha, \beta$-Crown. However, none of the *hard* networks from Wang et al. (2021) that we consider have been tested. We thus tested three representative *hard* DNNs (first 100 images) to understand how MN-BaB fairs on such hard instances, and report the numbers in Table 10. Results are directly comparable with Table 3.

Analysis: results reveal that MN-BaB is slightly slower and slightly less accurate than $\alpha, \beta$-Crown. Notice the specially high number of undecided images for CNN-B-Adv with TO=30s, probably meaning that 30s is too small for MN-BaB on this large DNN. Hence, Hybrid MILP is more accurate than MN-BaB with similar runtime or faster.

## COMPARISON $\alpha, \beta$-CROWN VS NNENUM

NNenum Bak (2021) is a complete verifier with good performance according to VNNcomp. It was the only complete verifier tested in Table 2 to verify more images than $\alpha, \beta$-Crown. The experiments section in Bak (2021) does not report the *hard* DNNs we are considering. We tried to experiment it on the same MNIST $6 \times 500$ and CIFAR CNN-B-adv as we did in Table 10 for MN-BaB. Unfortunately, on $6 \times 500$, buffer overflow were reported. We report in Table 11 experiments with the same

| Network | $\alpha, \beta$-Crown | Refined $\beta$-Crown | PRIMA |
|---|---|---|---|
| MNIST $5 \times 100$ | N/A | 14.3% (102s) | 33.2% (159s) |
| MNIST $5 \times 200$ | N/A | 13.7% (86s) | 21.1% (224s) |
| MNIST $8 \times 100$ | N/A | 20.0% (103s) | 39.2% (301s) |
| MNIST $8 \times 200$ | N/A | 17.6% (95s) | 28.7% (395s) |
| MNIST $6 \times 500$ | 51% (16s) | — | 64% (117s) |
| CIFAR CNN-B-adv | 18.5% (32s) | — | 27% (344s) |
| CIFAR ResNet | 0% (2s) | — | 0% (2s) |

Table 9: Undecided images (%, *lower is better*), as computed by $\alpha, \beta$-Crown, Refined $\beta$-Crown, and PRIMA, as reported in Wang et al. (2021), except for $6 \times 500$ that we run ourselves. N/A means that Wang et al. (2021) did not report the numbers, while $-$ means that Refined $\beta$-Crown cannot be run on these DNNs.

| Network | $\alpha, \beta$-Crown TO=30s | $\alpha, \beta$-Crown TO=2000s | MN-BaB TO=30s | MN-BaB TO=2000s |
|---|---|---|---|---|
| MNIST $5 \times 100$ | 55% (19s) | 50%(1026s) | 60% (19s) | 50% (1027s) |
| MNIST $6 \times 500$ | 51% (16s) | 50% (1002s) | 58% (18s) | 55% (1036s) |
| CIFAR CNN-B-adv | 22% (8.7s) | 20% (373s) | 43% (14s) | 24% (576s) |

Table 10: Undecided images (%, *lower is better*), as computed by $\alpha, \beta$-Crown, and MN-BaB

2000s Time-out (it was $10000s$ in Table 2) for a fair comparison with $\alpha, \beta$-Crown, on both MNIST $5 \times 100$ and CIFAR CNN-B-Adv. On MNIST $5 \times 100$, NNenum is slightly more accurate than $\alpha, \beta$-Crown, but far from the accuracy Hybrid MILP. On CIFAR CNN-B-adv, NNenum was much less accurate than $\alpha, \beta$-CROWN, and thus of Hybrid MILP. In both test, the runtime of NNenum was also much longer than for Hybrid MILP.

| Network | $\alpha, \beta$-Crown TO=2000s | NNenum TO=2000s | Hybrid MILP |
|---|---|---|---|
| MNIST $5 \times 100$ | 50%(1026s) | 44% (1046s) | **13% (46s)** |
| CIFAR CNN-B-adv | 20% (373s) | 40% (1020s) | **11% (417s)** |

Table 11: Undecided images (%, *lower is better*), as computed by $\alpha, \beta$-Crown and NNenum with 2000s time-out, and Hybrid MILP

.

# E   AVERAGE VS MAX TIME PER PMILP CALL

We provide in Table 12 the average as well as maximum time to perform $MILP_X$ calls as called by pMILP, on a given input: image 3 for MNIST, and image 76 for CIFAR10. For 6x500, we provide results for two different $\varepsilon$, following our test from Figure 3.

| Network | average time | maximum time |
|---|---|---|
| MNIST $5\times100$ $\epsilon = 0.026$ | 0.41s | 1.87 |
| MNIST $5\times200$ $\epsilon = 0.015$ | 0.75s | 5.31s |
| MNIST $8\times100$ $\epsilon = 0.026$ | 0.39s | 1.41s |
| MNIST $8\times200$ $\epsilon = 0.015$ | 0.49s | 1.63s |
| MNIST $6\times500$ $\epsilon = 0.035$ $\epsilon = 0.1$ | 1.4s 44.6s | 3.5s 310s |
| CIFAR CNN-B-adv $\epsilon = 2/255$ | 1s | 609s |

Table 12: average and maximum time per MILP$_X$ calls for image 3 (MNIST) and image 76 (CIFAR10).

Notice that DNN $6\times 500$ and $\epsilon = 0.1$ is a very hard instance as being very close to the falsification $\epsilon \approx 0.11$. This is thus not representative of the average case. Also, on this image 3, pMILP succeeds to verify $\epsilon = 1.054$, while $\alpha, \beta$-CROWN can only certify $\epsilon = 0.0467$ within the 10 000s Time-out.

For CNN-B-Adv, the very long maximum time for a MILP call is an outlier: it happens only for one output layer, for which the number $K$ of open nodes is particularly large (around 200 out of 20000 neurons) to certify this hard image 76. Indeed, the average time is at $1s$. Notice that this does not lead to a runtime of 20.000s, as 20 threads are used by pMILP in parallel (similar to competing solutions, except $\alpha, \beta$-CROWN which uses $> 4096$ GPU cores).