# OpenReview forum: "Hybrid MILP to efficiently and accuratly  solve hard DNN verification instances"
_ICLR.cc/2025/Conference — Submitted to ICLR 2025_

### Official Review · Reviewer_zhoq · 2024-10-27

**Soundness:** 2
**Presentation:** 1
**Contribution:** 3
**Rating:** 6
**Confidence:** 5

**Summary:**

This work proposed Hybrid MILP, a neural network verifier which first uses the optimisation based ab-CROWN to solve easy instances before invoking a MILP solver to solve the remaining hard instances. They key to Hybrid MILP is to only encode a subset of unstable neurons using binary variables (i.e., to 'open' them). This subset is chosen based on an upper bound on the individual neurons utility in such an encoding. The resulting method demonstrates strong performance on hard verification instances on small DNNs.

**Strengths:**

* Novel utility function based on the primal pre-activation values of neurons instead of just their bounds.
* Empirically high effectiveness across a range of neural networks
* Extensive background and discussion of (some) related work.

**Weaknesses:**

* Lack of ablation studies confirming the importance of the proposed utility function and partial encoding (see below).
* Key parameters of the experimental setup for the main experiment in Table 4 are not discussed, e.g., how was K chosen, how were intermediate bounds computed, how was 'z' chosen for the utility function computation, were different sets used for robustness against different alternative classes
* Lack of theoretical and empirical, comparison (or even mention) of the closely related work on branching heuristics (e.g. Bunel et al, De Palma et al., Ferrari et al. and Henriksen et al.) , similarly trying to estimate the importance of encoding neurons exactly in BaB.
* Lack of comparison to the optimizer's relaxation strategy at equal runtime, i.e., the importance of partial encodings in the first place.
* Applicability only to very small DNNs, which previous work (Ferrari et al.) found to be more easily solvable by LP/MILP based verifiers like ERAN, which were not compared to.
* Overclaims regarding the novelty of Proposition 1, which was discussed in similar form in Singh et al Equation 2 and Salman et al. Theorem 4.2.
* Poor copywriting and large number of typos, including in and the abstract formulas (e.g. Line 174 (half sentence missing), Line 292 (UB - UB -> UB - LB), first expression in Line 359 (three closing but only one opening bracket))

**Minor Comments**
* BaB-based methods such as ab-CROWN and MN-BaB are complete, but the opposite is implied in several places.
* Completeness is binary and can not be traded of with efficiency, but precision (at a given timeout) can.

**References**
* Ferrari et al. "Complete verification via multi-neuron relaxation guided branch-and-bound."
* Salman, et al. "A convex relaxation barrier to tight robustness verification of neural networks."
* Singh et al. "An abstract domain for certifying neural networks."
* Bunel et al. "Branch and bound for piecewise linear neural network verification."
* De Palma et al. "Improved branch and bound for neural network verification via lagrangian decomposition."
* Henriksen et al. "DEEPSPLIT: An Efficient Splitting Method for Neural Network Verification via Indirect Effect Analysis."

**Questions:**

1) Can you include experiments using a full-MILP encoding and different partial MILP encodings using other utility functions after ab-CROWN to better assess the contribution of the novel utility function vs. the combination of ab-CROWN with (any) MILP based strategy?
2) How does your partial MILP encoding compare to a full MILP encoding (of unstable ReLUs) in terms of the achieved upper bounds over time? This is particularly interesting, as a popular approach to solving MILP problems is exactly to relax binary variables (automatically and based on the solvers strategies).
3) How is the solve time in Table 3 affected by the choice of neurons? And how do the results change if no exact MILP bounds but only LP bounds are available for Layer 2?
4) Can you discuss the experimental details as per the weaknesses?
5) Can you include a formulation of the inductive definition of the utility function?
6) Can you include experiments on larger networks to investigate the limitations of the proposed method?

**Conclusion**
The work is a promising new direction for solving hard verification instances on small Networks. However, comparison to key related work is missing and experimental validation is too granular to assess the effectiveness of the novel components proposed in this work (see weaknesses). In addition, the limitations of this work with regards to applicability to larger and perhaps more relevant networks are not investigated. Overall, I believe this work does thus not meet the bar for acceptance at ICLR but I am more than happy to reconsider this assessment should my concerns be addressed.

---

> ### Author Response · Authors · 2024-11-12
>
> Thanks for your very detailled review.
>
> Some answers now to the questions (which are already in the paper), more later:
>
> >Q1: Can you include experiments using a full-MILP encoding
>
> full MILP is well known not to scale. Still, we reported an experiment on the smallest of the network (6x100): see Table 2 page 2 "full MILP", realized with a very long 10.000s per image: 50% undecided inputs (90%-40%) for average 6100s per image, compared with 13% undecided images and 46s using Hybrid MILP. And this is the smallest network.
>  The analysis is stoppedf for each output neuron (which is not the predicted class) as soon as we have ceritification  or falsitication of robustness, which is the most efficient complete test. Do you think it is necesarry to perform this test for larger networks? The outcome is extremely obvious.
>
> > and different partial MILP encodings using other utility functions
>
> the only other utility function proposed was by Huang et al. 2020. Table 3 page 8 gives actually a more precise picture that just raw % of robustness, by looking at precise accuracy results. Again, we can do it on say 6x100, but the outcome is also extremely obvious (accuracy number from alpha beta CROWN alone, because the accuracy of pMILP with other utility function and the same number of nodes is very poor). Unless we push the number of nodes of pMILP with other utility function to much higher number, but then, the runtime will be extremely high. We will provide both results to give an exact picture. But again, the overall picture is already very clear from Table 3.
>
>
> >Q2 How does your partial MILP encoding compare to a full MILP encoding (of unstable ReLUs) in terms of the achieved upper bounds over time? This is particularly interesting, as a popular approach to solving MILP problems is exactly to relax binary variables (automatically and based on the solvers strategies).
>
> Isnt Table 3 page 8 answering this question as well?
> The max line corresponds to full MILP. Specifically, Read line 420 and 421: opening 35 neurons with our utility function reaches 95% of the accuracy of full MILP (all 116 unstable nodes open).
>
> >Q3 How is the solve time in Table 3 affected by the choice of neurons?
>
> We will report that later. The complexity analysis tells us that it is exponential in the number of chosen neurons.
>
> > And how do the results change if no exact MILP bounds but only LP bounds are available for Layer 2?
>
> We will report such numbers. We expect the number of open nodes in layer 3 has to be considerably higher to reach the same accuracy. Notice that computing bounds in layer 2 is fast (even full MILP can actually be done for layer 2 in short time).
> In our prototype, we open 48 nodes for layer 2, and the accuracy is >99.99% of full MILP. But we understand it as ablation study to  see the impact of it on accuracy.
>
>
> >Q4 Can you discuss the experimental details as per the weaknesses?
> "Key parameters of the experimental setup for the main experiment in Table 4 are not discussed"
>
> Well, this is provided in Table 5, page 13 (appendix).
>
> >Q5 Can you include a formulation of the inductive definition of the utility function?
>
> Do you mean for more than 2 layers back?
>
> >Q6 Can you include experiments on larger networks to investigate the limitations of the proposed method?
>
> CNN-B-Adv has ~20.000 neurons, CIFAR Resnet has almost ~100,000 neurons.
> We will certainly not scale past what alpha, beta Crown is capable of.
> Our method is meant to be more accurate than alpha beta Crown, not faster or more scalable.

---

> > ### Comment · Reviewer_zhoq · 2024-11-13
> >
> > I thank the authors for their quick reply. However, it unfortunately does not address the majority of my questions. In particular:
> >
> > Q1) The key part of my question is *after ab-CROWN*. I.e. how does pMILP compare to full MILP (and other utility functions) when run after ab-CROWN using the resulting relatively tight intermediate bounds.
> >
> > Q2) No, Table 3 does not address this question, as solve time and number of encoded neurons are only loosely correlated for real solvers (some neurons are very easy to stabilize via cutting planes). A "cactus plot" would be the best way to visualize this.
> >
> > Q4) While Table 5 provides details on the number of open neurons and timeouts, ab-CROWN has a large number of further parameters that are not reported.
> >
> > Q5) A single equation containing the whole (inductive) definition of the utility function instead of individual terms distributed across half a page of writing would improve clarity.
> >
> > Q6) The ResNet is irrelevant given its training and poor accuracy and 20k neurons is quite small for recent verification methods. The CIFAR-10 networks from Ferrari et al. could be interesting.

---

> ### Author Response · Authors · 2024-11-13
>
> >Q1) The key part of my question is after ab-CROWN. I.e. how does pMILP compare to full MILP (and other utility functions) when run after ab-CROWN using the resulting relatively tight intermediate bounds.
>
> Two answers:
> a) we do not use any intermediate bounds computed from ab-CROWN when running pMILP. This might be doable, but would need a low level integration. In our prototype, ab-CROWN is only used to solve easy instances. Hard instances (that is, instances for which abCROWN fails to answer) are restarted from scratch. This is already very efficient, although it could be made even more efficient by using intermediate abCROWN bounds.
>
> In this respect, on 6x100, full MILP did not solve any of the hard instances not solved by a fast run of ab-CROWN, although the time out was 10.000s.
>
> b) We tried 6x100 with computing intermediate bounds using full MILP (iterative full MILP).
> Some hard instances not solved by abCROWN can be solved by this iterative full MILP, but the runtime to solve such instances is 20x higher than what is achieved by pMILP, without solving more images than pMILP.
>
> > Q2:
> we have run the numbers. Scroll down to see the result (no space to input the full result here)
>
>
>
> > Q4) While Table 5 provides details on the number of open neurons and timeouts, ab-CROWN has a large number of further parameters that are not reported.
>
> The networks we experiment with have already been tested by abCROWN.
> We thus use the configuration files which come for each network in abCROWN (except for time-out which we explicitly mention).
>
> e.g.: for CNN-B-Adv:
> https://github.com/Verified-Intelligence/alpha-beta-CROWN/blob/main/complete_verifier/exp_configs/beta_crown/cifar_cnn_b_adv.yaml
> "solver:
>   batch_size: 512
>   beta-crown:
>     iteration: 20"
>
> and
>
> for 6x100:
> https://github.com/Verified-Intelligence/alpha-beta-CROWN/blob/main/complete_verifier/exp_configs/beta_crown/mnist_6_100.yaml
>
> "solver:
>   batch_size: 1024
>   beta-crown:
>     iteration: 20"
>
> Our experimental results are in line with published numbers.
>
> > Q5) A single equation containing the whole (inductive) definition of the utility function instead of individual terms distributed across half a page of writing would improve clarity.
>
> Understood. We will provide a simpler form definition.
>
> > Q6) The ResNet is irrelevant given its training and poor accuracy and 20k neurons is quite small for recent verification methods.
>
> Number of neurons is not the only parmeter to take into account. What you are saying is true for EASY instances. For hard instances, as we shown, even small networks cannot be tackled fully by current verification methods.
>
> Our tool scales to  at least 20k neurons.
>
> > "20k neurons is quite small for recent verification methods. The CIFAR-10 networks from Ferrari et al. could be interesting."
>
> From Ferrari et al,  2 CIFAR10 Networks stand out as being "harder to verify":
> ResNet6A  (20% undecided images) and ResNet8A (47% undecided images), Table 1 page 7 of https://arxiv.org/pdf/2205.00263 .
>
> For other CIFAR10 networks in Ferrari et al., current methods are efficient enough(<4% undecided images), and better accuracy wont tremendously change the picture.
>
> These 2 networks ResNet6A and ResNet8A actually have LESS neurons (<12k, Table 3 page 14 of https://arxiv.org/pdf/2205.00263 ) than CNN-B-Adv (~20k), so I do not really understand the comment: "20k neurons is quite small for recent verification methods."
>
> Thanks for pointing out more <20k neurons networks which are not easy to verify, this actually makes our point stronger.
>
> That being said, I perfectly understand where you come from, as I had the same belief before working on this paper:
> "current verification tools are fine up to ~100 000 neurons." I changed my mind. One of the paper contribution is actually to expose that fact.
>
> The fact is, we tried to be as fair as possible (by using networks where abCROWN has already been tested, with configuration files provided by the abCROWN authors, with results matching the published results), and what we found suprised at least ourselves.

---

> ### Author Response · Authors · 2024-11-14
> **Proposition 1**
>
> >	Proposition 1 is established in prior works.
>
> We are looking at each of the 4 papers mentioned by the reviewers on the issue.
>
> [A] “Formal Verification of Piece-Wise Linear Feed-Forward Neural Networks”, ATVA, 2017.
>
> [B] “Input-Relational Verification of Deep Neural Networks”, PLDI, 2024.
>
> [C] Singh et al. "An abstract domain for certifying neural networks" Equation 2.
>
> [D] Salman, et al. "A convex relaxation barrier to tight robustness verification of neural networks. 2019". Theorem 4.2
>
>
> We did not find a clear and explicit statement which would be equivalent with our Proposition 1. For papers [B] and [D], we are exploring further to see if their results (specifically Theorem 4.2) would imply Proposition 1 or if they would suggest in writing something similar to Proposition 1. For two papers: [A] and [C], the work are clearly orthogonal to our statement.
>
> We believe there might be a misunderstanding in what our statement claims, which would explain why [A] and [C] at least are suggested as establishing the results by the reviewers.
>
> To rephrase in English, Proposition 1 claims that the LP relaxation (of the standard exact MILP encoding of ReLU) is EQUIVALENT with the triangle approximation. And [A;C] are simply NOT dealing with LP relaxation!
>
> To say differently, Proposition 1 does not claim the triangle approximation to be novel – it certainly isn’t ([A],[B],[C],[D] are non exhaustive examples of that). But the EQUIVALENCE may be new, and at least it is not that obvious or well-known. That being said, it might already be proved in another paper ([D] or somewhere else). The proof is not that complicated, but this is certainly very important and instrumental in the way our Utility function is so much more efficient than what was proposed before (more on the utility function later). If Proposition 1 is already proved somewhere else, then it does not impact our paper much.
>
> We certainly agree that the idea of a triangle approximation appeared countless time before (in these 4 papers specifically, but in numerous other as well). We actually note such a fact immediately after Proposition 1, line 296 “Notice that this is close to the DeepPoly abstractions Singh et al. (2019b)…”

---

> ### Author Response · Authors · 2024-11-14
> **Utility Function vs branching heuristics**
>
> > Utility functions, in principle, appear similar to branching heuristics (e.g., BaBSR).
>
>
> After investigating several papers, we can say the following:
>
> Most (if not all) heuristics in DNN verification consider how each (uncertain) neuron affects a certain node X (e.g. an output neuron X): e.g. BaBSR, Huang et al. 2020, etc.
>
> Our Utility function is no different in that respect: it ranks neurons by how important they are to a given node X, to only consider the most important neurons accurately.
>
> Similarities stop there with BaBSR and previous heuristics.
>
> Now, the key difference, and the reason why our Utility is so efficient (it can even make the usually very inefficient MILP solvers more efficient than the otherwise state of the art BaB solvers. Compared to Huang et al. 2020, it needs to open 4 times less neurons for the same accuracy) is the following.
>
> It uses the solution to an LP call of a min/max query on node X (as computed by Gurobi) (line 312). As it is LP, the solution computed is also the bound in the dual space. The cost of this LP call is negligeable wrt the forthcoming call to pMILP, and thus the time taken here is certainly worth.
>
> Now, we use this solution, which associates each variable/neuron with a specific value, to have a much more accurate understanding of how each variable/neuron affects X. That’s where Proposition 1 is crucial, in telling us how much accuracy is gained by opening that neuron vs LP relaxation, and we can rank them extremely accurately (Table 3 is extremely clear in that respect). Proposition 2 is interesting, but is far from capturing how close UTILITY is to IMPROVEMENT (in our test, we were 99% close- of course this cannot be proven in general as we could manually generate degenerated cases where the distance is much larger).
>
> Compared to e.g. Huang et al 2020, it allows to have a very precise answer *locally* for that input, while Huang et al. considers mostly the DNN, and less the input. This is also what happens with BaBSR, although comparing pMILP and BaB cannot be made very directly.
>
> To the best of our knowledge, this is the first time such a solution (through an LP call) is used to refine the contribution of each neuron to a given node X.

---

> ### Author Response · Authors · 2024-11-15
> **Q2: time and accuracy for different number of nodes open**
>
> > Q2
>
> Here are the results we have on pMILP with different number of open nodes, and time to compute the full 100 nodes of layer 3:
> Notice that the image chosen (59) is particularly hard (because there are lots of uncertain nodes), hence the quite large time to compute the layer.
>
>
> This is the same test with comparable results as in Table 3, with previous layer L2 computed using full MILP.
>
>  nbr          ..            time    .................                                   acc
>
> 1	.. 5.992132187  ...	1.702986873
>
> 2 ..	9.024885654 ...	1.654690348
>
> 3 ..	10.6186769 ...	1.612137282
>
> 4 ..	11.60420394 ...	1.571001109
>
> 5 ..	12.92842579 ...	1.531925404
>
> 6  ..	12.98235846 ...	1.49535638
>
> 7 ..	13.66908836 ...	1.46189314
>
> 8 ..	14.47656775 ...	1.429953534
>
> 9 ..	15.1073184 ...	1.400636414
>
> 10 ..	16.18551326 ...	1.37112038
>
> 11 ..	17.52154756 ...	1.343824578
>
> 12 ..	17.83653903 ...	1.31833565
>
> 13 ..	18.52695084 ...	1.293869092
>
> 14 ..	19.86539316 ...	1.2690508
>
> 15 ..	21.07112885 ...	1.247510672
>
> 16 ..	23.13534665 ...	1.224306561
>
> 17 ..	 25.1482892 ...	1.203179112
>
> 18 ..	27.09270978 ...	1.183947413
>
> 19 ..	29.32548857 ...	1.164465319
>
> 20 ..	33.14418769 ...	1.145618187
>
> 21 ..	40.39810514 ...	1.126125249
>
> 22 ..	45.63475466 ....	1.108974532
>
> 23 ..	53.25520349 ...	1.093124202
>
> 24 ..	62.35421562 ...	1.07730883
>
> 25 ..	101.9047432 ...	1.060928575
>
> 26 ..	113.4482219 ...	1.045715429
>
> 27 ..	154.7245564 ...	1.030605317
>
> 28 ..	196.723217 ...	1.016058572
>
> 29 ..	228.8618577  ...	1.00108374
>
> 30 ..	279.520607 ...	0.987652522
>
> max=116 .. 3300 ... 0.895
>
> we can clearly see the exponential scaling with the number of open nodes, taking more and more time for each additional open node. On this layer, our standard pMILP prototype will open 21 nodes (cf Table 5).
> This takes 100 times less than using full MILP (cf max = 116 nodes), while providing 85% of the accuracy difference between 0 to 116 open nodes.
>
> > If Layer 2 is done with LP instead of full MILP, the accuracy of layer 3 is way worse:
>
> nbr .. time ..............acc with L2 LP ....acc with L2 with fullMILP for reference
>
> 5	.. 9.362099648 ...	3.247378853 ...	1.53
>
> 10	.. 10.64601064 ...	3.022145891 ...	1.37
>
> 15	.. 11.97100925 ...	2.823833569 ...	1.25
>
> 20	.. 13.12460947 ...	2.638627159 ...	1.15
>
> 25	.. 16.02304268 ...	2.47324218 ...	1.06
>
> 30	.. 28.33429575 ...	2.327935601 ...	0.99
>
> 35	.. 48.17112136 ...	2.195061671
>
> 40	.. 89.42136264 ...	2.071075926
>
>
> This shows how important it is to compute quite accurately layer by layer.
> The exponential complexity in the number of open nodes is also very clear.
> time per similar number of nodes is better compared with the previous test, but time per similar accuracy is way worse (which is what matters).

---

> ### Author Response · Authors · 2024-11-15
> **pMILP vs inductive full MILP**
>
> We did another interesting test to highlight why chosing nodes to open is so important.
> It is not that important for layer 3, full MILP is runnable (although it takes much longer than pMILP for a very negligeable improved accuracy).
>
>
> More interesting is what happens in layer 5 (last before output), where using full MILP would be a very bad idea, even for the smallest 5x100 DNN.
>
>
> We run pMILP to compute the bounds for the first 4 layers. We use these bounds to run pMILP vs full MILP on layer 5.
> The results are as follows:
>
>
>  for pMILP:
>
> nbr..time .... acc
>
> 1 ..14 .. 3.233021901
>
> 2 ..15.3 ..	3.140309921
>
> 3 .. 17.21 ..	3.059083103
>
> 4	.. 17.4 ..	2.986166762
>
> 5 .. 19.2 ..	2.856229765
>
> 6 .. 20.95 ..	2.799248232
>
> 7 .. 23.7 ..	2.746167245
>
> 8 .. 26.67 .. 2.69485246
>
> Now, if we run full MILP (with the many undecided nodes of image 59), here is what we get by using different time out settings:
>
> time ... acc
>
> 21.1	... 3.348236261
>
> 27.6	... 3.24604282
>
> 38.2 ...	3.196640184
>
> 47.1	... 3.164298172
>
> 56.69 ... 3.146913614
>
> 106.7 ...	3.108035223
>
> 156.32 ...	2.900438725
>
> 406.7 ... 2.800268264
>
> 606.1 .. 2.737064255
>
> This shows that chosing nodes is much more efficient for time/accuracy trade-off than using full MILP.
> Helping Gurobi with the nodes to consider is extremely effective:
>
> pMILP obtains 2.8 accuracy in <21 seconds (with 7 open nodes), while full MILP needs 400 seconds to obtain it, a 19x speed up.
>
> for 2.7 accuracy, the speedup is >>22.
>
> And this is for the smallest DNN we considered (500 hidden neurons, far from the biggest 20k neuron DNN we experimented with)

---

> ### Author Response · Authors · 2024-11-20
>
> Dear Reviewer,
>
> We have updated the pdf paper with the ablation study and the formula in the appendix (p15+), with graphs easier to parse than the table in comments (see blue). We also rewrote the introduction, in particular our main contributions (in blue as well).
>
> We will update later the core of the paper with the novelty of our Utility function as well as more discussion on Proposition 1, as we discussed in comments above, and to integrate an easier to parse Utility formula.
>
> By the way, we tried to locate Resnet8-A (which is the most interesting of the ones you suggested, as it displays the largest number of undecided images).
>
> We checked the github
> https://github.com/eth-sri/mn-bab/tree/SABR_ready/networks
>
> we could find resnet_2b and resnet_4b_bn_adv_2.2_bs256_lr-3.pth.
>
> We tried to match them with Resnet6A and Resnet8A.
> The numbers are quite different with Ferrari et al. 2022 though:
>
> resnet_2b  has abCROWN verifies 67% of images, with 0% undecided images.
>
> resnet_4b_bn  has abCROWN verifies 32% of images, with 41% undecided images (upper bound is at 73%).
>
> Also, resnet_4b_bn has 14.4K activation neurons, different from the <12K from Resnet-6A and Resnet-8A.
>
> We are investigating the resnet_4b_bn network (14.4K activation neurons), which seems interesting anyway.

---

> ### Author Response · Authors · 2024-11-21
> **New revision of the pdf**
>
> Dear Reviewers,
>
> Thanks again for your very constructive guidance.
>
> We have updated the pdf in 2 key parts (Section 3 and 4, see the blue), answering the misunderstanding on the reviewers major concerns:
>
> 1) on Proposition 1 and the triangular abstraction, we introduce the triangular abstraction earlier, with citations, and state Proposition 1 with an equivalent but clearer and easier to parse statement using the triangular abstraction,
>
> 'the LP relaxation of the exact MILP encoding is equivalent with the triangular abstraction"
>
> plus a note that to the best of our knowledge, such an equivalence  statement cannot be found in the bibiliography. This is also reflected in the introduction.
>
> 2) on the noveltty of  our Utility function and comparison with other ranking functions, we rewrote Section 4 entirely, stating the key novelty (the fact that we use a (single) call to LP solver in order to rank nodes, which is novel as far as we know), as well as provide a clearer and shorter definition of the Utility function. This is also reflected in the introduction.
>
> again, the  ablation study from the discussion can be found in the appendix, together with illustrative graphs, showing that each feature brings valuable improvement (2x on accuracy for every layer by using incremental computation layer by layer, 20x speed up of pMILP vs MILP with time-outs).

---

> ### Comment · Reviewer_zhoq · 2024-11-24
>
> I thank the authors for the additional clarifications and results.
>
> I still believe that Proposition 1 is a weaker form of e.g. Corollary 4.3 from Salman et al. and widely known in the field. I would strongly encourage the authors to drop their claim of novelty or can not recommend acceptance.
>
> In light of the additional results and improved write-up, I agree with the authors that the proposed method is promising for the class of networks investigated here and can outperform existing methods by a notable margin and will thus raise my score.
>
> I however also strongly encourage the authors to improve the presentation of this work. This includes:
> * A more thorough (empirical and conceptual) comparison to the heuristics proposed in prior work (using the same underlying bounds to evaluate their scores)
> * Improving the quality (and selection) of plots and tables presented in this work (e.g. why is Table 7 not a plot or at least has an extra column?, Why not present Table 3 as a plot? Why use a line instead of a scatter plot for Figure 2? Axis labels are typically too small)
> * Further improvements of Section 4 highlighting the use of the primal solution not available in ab-CROWN.

---

> ### Author Response · Authors · 2024-11-25
>
> Thanks for the comments.
>
> We are implementing the changes you are suggesting and will commit ASAP.
>
> On proposition 1: We take your word that this is known in the field.
>
> edit:
> We have pushed a new version of the pdf, with changes in the abstract, introduction (main contributions) and chapter 3, line 279/280:
>
> > we invoke a folklore result on the LP relaxation of (2), for which we provide a direct and explicit proof:
>
> We believe the changes will  provide a consensual picture of the matter on Proposition 1.
>
> On a side note, rereading Salman et al., we understand the following:
>
> - They state page 4 that easily, the *convex relaxation* of the feasible set of MILP(ReLU) is the triangular abstraction (=from the primal view). That's obvious.
>
> Now, what I am not sure about:
>
> Is the *convex* relaxation of the feasible set of a MILP problem necessarily the same in general as the feasible set of the *linear* relaxation of the MILP problem?
>
> One inclusion is obvious as the feasible sets of Linear problems are convex, but the other inclusion does not seem immediate to me. It may be known to be true in all cases, in which case the proof of Proposition 1 is obvious relying on that result. If not, then I do think Proposition 1 needs a proof (even though folklore).
>
> Concerning Theorem 4.2 and Corrolary 4.3, they state that the bound of the *convex* relaxation of the primal view (here triangular inequality) are the same as the bound of the *convex* relaxation of the dual view. We could not find a connection  between the dual view and the standard MILP encoding of ReLUs.
>
> Best,

---

> ### Author Response · Authors · 2024-11-26
> **New PDF available**
>
> A new PDF version is available.
>
>
> The presentation of Proposition 1 should be in a consensual state now.
>
> We have remade all the graphs plus additional ones.
>
>
> On top of that:
>
> >A more thorough (empirical and conceptual) comparison to the heuristics proposed in prior work (using the same underlying bounds to evaluate their scores), and
>
> >Further improvements of Section 4 highlighting the use of the primal solution not available in ab-CROWN.
>
> -Conceptually, we added yet another explanation and comparison with usual heuristic on page 6:
>
>
> >Usually, heuristics to choose X are based on evaluating the sensitivity of a neuron z wrt the ReLU
> nodes, that is how much a ReLU value impacts the value of z, and rank the ReLU nodes accordingly.
> This is the case of Huang et al. (2020), but also more generally of heuristics for BaB Bunel et al.
> (2020); Ferrari et al. (2022), such as SR and FSB. Instead, Utility considers the improvement from
> opening a neuron n, that is the difference for the value of z between considering ReLU(n) exactly
> or using its LP relaxation LP(n). Indeed, it is not rare that z is sensitive to ReLU node n, and yet
> LP(n) already provides an accurate approximation of ReLU(n). In this case, usual heuristics would
> open n, while it would only improve the value of z in a limited way.
>
> We also highlighted many cases where indeed LP(n) IS accurately representing ReLU(n) (and thus usual heuristic based on sensitivity would wrongly open these nodes), explaining in part why Utility is so efficient.
>
> > Consider b with Wbz < 0: to maximize z, the value of sol(ˆb) is minimized, which is sol(ˆb) =
> ReLU(sol(b)) thanks to Proposition 1. Even if z is sensitive to this ReLU b, the improvement of b
> is 0. Utility does not open it as Utility max(b) = 0, whereas usual heuristics would.
>
>
>
> - Experimentally, we tested the SR and FSB heuristic used in BaB as heuristic to choose ReLU nodes to open.
>
>
> EDIT: We found a bug in our implementation of SR and FSB and corrected it, creating the graph page 16. Now, SR and FSB are significantly better than random, but still behind (Huang). We also tested with nodes chosen a single layer earlier, and the performance of SR and FSB are slightly worse than with several layers - which is normal and rules out a problem with ranking ReLU nodes in different layers.
>
>
> The table can be found page 16. The present results are not good for SR and even slightly worse for FSB, slightly worse than
> (Huang et al.) while SR and FSB can choose nodes several layer before and (Huang et al.) is stuck with nodes from the previous layer. We believe that:
>
> The objective to branch on BaB nodes, although close to opening ReLUs, is however different, and the heuristic is not adapted.
>
> Notice that we can easily replace Figure 2 page 8 by Figure 5 page 16. We would have done it already if SR or FSB was producing better choice for opening ReLU nodes than (Huang). As it is not the case, we do not think it is as importantn and  Figure 2 has less clutter than Figure 5.
>
> > Improving the quality (and selection) of plots and tables presented in this work
>
> We have reworked all the graphs and added some new ones.

---

### Official Review · Reviewer_ZYJY · 2024-11-03

**Soundness:** 3
**Presentation:** 3
**Contribution:** 3
**Rating:** 5
**Confidence:** 4

**Summary:**

This paper introduces Hybrid MILP, a method designed to efficiently solve complex verification problems in deep neural networks (DNNs), particularly those involving ReLU-based architectures. The paper addresses the limitations of the current state-of-the-art verifier, α, β-CROWN, which performs well on relatively simple instances but struggles with more challenging ones. The authors propose a hybrid approach that initially applies α, β-CROWN with a short time-out to quickly address easier instances. For instances that remain undecided, Hybrid MILP selectively applies a partial MILP (Mixed Integer Linear Programming) that combines integer variables with linear relaxations for a subset of neurons, reducing the number of integer variables by approximately four times compared to prior methods. This approach effectively narrows the undecided instance rate and is experimentally validated to be both accurate and efficient. Results on benchmark datasets, including MNIST and CIFAR, demonstrate Hybrid MILP’s effectiveness in reducing undecided cases by up to 43%, with a manageable runtime.

**Strengths:**

-  Hybrid MILP innovatively combines MILP and linear relaxation techniques to handle challenging verification tasks effectively.
- Experimental results highlight substantial improvements in both verification accuracy and runtime efficiency.

**Weaknesses:**

- No complexity analysis of the proposed method.

- Limited to ReLU activation functions.

**Questions:**

- Could the proposed method extend to MaxPool nonlinear layer? If can, how is the performance of Hybrid MILP compared to $\alpha,\beta$-CROWN?
- MILP is complete but time-consuming. I am curious about the complexity of the hybrid MILP and why it is more efficient than other baseline?

**Details Of Ethics Concerns:**

/

---

> ### Author Response · Authors · 2024-11-12
>
> Thanks a lot for your  comments.
>
> 1) MaxPool can be treated using MILP, see (Huang et al. 2020). The coding is however more complex, and I doubt we will have time to implement that during the short discussion period.
>
> 2) Concerning complexity analysis of pMILP, it is as follows:
>
> O(N · MILP(N, K)), where N is the number of nodes of the DNN, and MILP(N, K) is the complexity of solving a MILP program with
> K binary varialbes and N-K linear variables. We have that MILP(N, K) < 2^K LP(N), where LP(N) is the complexity of running LP, polynomial in N, thus the exponential is only on K. The key factor is thus to keep K as small as possible, which our Utility function enables (see Table 3).
>
> This explains why pMILP is so efficient.
>
> For reference, here is the pseudo code the analysis is based on.
> HybridMILP pseudo-code:
>
>     Run Alpha-Beta Crown with small time out (e.g. 10s).
>
>     If image undecided then run pMILP.
>
> pMILP code:
>
> Input: Bounds [α_m, β_m] for input nodes m at layer 0 (input neighbourhood)
>
> Output: Bounds [α_n, β_n] for every output node n
>
> for layer k = 1 · · · K do
>
>        for neuron n in layer k do
>
>            Compute the set Z of the K most important nodes before n according to Utility(n)
>
>           Run partial MILP(Z as binary, nodes / Z as linear variables, with constraints every node m<n satisfy x_m \in [α_m, β_m]  )
>          to obtain bounds [α_n, β_n] for node n.
>
> return Bounds [α_n, β_n] for every output node n

---

> ### Author Response · Authors · 2024-11-21
> **reason for efficiency**
>
> > . I am curious about the complexity of the hybrid MILP and why it is more efficient than other baseline?
>
>
> The reason for efficiency lies in our Utility function: the reason why our Utility is so efficient (it can even make the usually very inefficient MILP solvers more efficient than the otherwise state of the art BaB solvers. Compared to Huang et al. 2020, it needs to open 4 times less neurons for the same accuracy) is the following.
>
> It uses the solution to an LP call of a min/max query on node X (as computed by Gurobi) (line 312). As it is LP, the solution computed is also the bound in the dual space. The cost of this LP call is negligeable wrt the forthcoming call to pMILP, and thus the time taken here is certainly worth.
>
> Now, we use this solution, which associates each variable/neuron with a specific value, to have a much more accurate understanding of how each variable/neuron affects X. That’s where Proposition 1 is crucial, in telling us how much accuracy is gained by opening that neuron vs LP relaxation, and we can rank them extremely accurately (Table 3 is extremely clear in that respect). Proposition 2 is interesting, but is far from capturing how close UTILITY is to IMPROVEMENT (in our test, we were 99% close- of course this cannot be proven in general as we could manually generate degenerated cases where the distance is much larger).
>
> Compared to e.g. Huang et al 2020, it allows to have a very precise answer locally for that input, while Huang et al. considers mostly the DNN, and less the input. This is also what happens with BaBSR, although comparing pMILP and BaB cannot be made very directly.
>
> To the best of our knowledge, this is the first time such a solution (through an LP call) is used to refine the contribution of each neuron to a given node X.

---

> ### Author Response · Authors · 2024-11-27
>
> Dear Reviewer ZYJY,
>
> Could you please find some time to look at either our rebutal, and/or the heavily reworked pdf (with blue highlights)?
>
> We believe that we answered most of your concerns.

---

> > ### Comment · Reviewer_ZYJY · 2024-12-02
> >
> > Thank you for your response. While I understand that obtaining CPLEX might be challenging for you, I still believe it’s necessary to compare with GCP-CROWN, as it has shown better performance than alpha-beta-CROWN. Perhaps you could directly use the results reported for GCP-CROWN and compare against them.

---

> ### Author Response · Authors · 2024-12-02
>
> Dear Reviewer,
>
> Sure, we can add the results reported for GCP-CROWN in the appendix, as we did for PRIMA.
> Notice that we cannot update the pdf anymore during the discussion period, but we will do it for the final version.
>
> As we said, GCP-CROWN is reported to be ~2% more accurate than ab CROWN on CNN-B-Adv.

---

> > ### Comment · Reviewer_ZYJY · 2024-12-02
> >
> > Dear authors,
> >
> > Thank you for your detailed work and clarifications so far. I was wondering, why didn’t you use the models from Table 4 of the GCP-CROWN paper? The activation functions in those models are also ReLU, which should align well with your approach. Could you provide some insights into this choice?

---

> ### Author Response · Authors · 2024-12-02
>
> Dear Reviewer,
>
> We believe that you are talking about Table 3 page 9 of
> https://arxiv.org/pdf/2208.05740
>
> Most of the networks here have small number of undecided images.... except for CNN-B-Adv >15%, which we do test.
> The second with most undecided images is CNN-A-Mix with 5.5% of undecided images.
>
> As time and computational ressources are always limited, we focused on DNNs with high number of undecided images, for which our accurate procedure could show a strong improvement.
>
> On CNN-A-Mix, it would be impossible to reach more than 5% improvements, and even less on other networks, the time would thus not be worth investing.
>
> Also, remember that Hybrid MILP first call abCROWN (but it could be GCP-CROWN when we will ave access to CPLEX) so all the accuracy and runtime would be carried over - see the case of Resnet which shows virtually no difference. In such DNN (except CNN-B-Adv), pMILP is simply not required.

---

> > ### Comment · Reviewer_ZYJY · 2024-12-02
> >
> > Dear Authors,
> >
> > Thank you for your response. I understand your reasoning regarding the #undecided images. However, I noticed that Table 4 in the appendix includes nine models, yet it seems Hybrid MILP was only applied to one of them. Was this decision entirely based on the #undecided images?
> >
> > Additionally, while I understand that extending your method to MaxPool layers might take additional effort, MaxPool is a very common layer, and MILP itself can naturally handle MaxPool operations (as shown in Huang et al., 2020). Moreover, open-sourced benchmarks with MaxPool layers are provided in VNN-COMP 2021 and 2023. Given this, Hybrid MILP should ideally be capable of addressing models with MaxPool layers, rather than being limited to ReLU-only models.  Improvements on MaxPool would make the method even more versatile and applicable to a wider range of real-world benchmarks.

---

> ### Author Response · Authors · 2024-12-02
>
> First, let me recap the discussion with the other 2 reviewers.
> The discussions revolved mainly around the heuristic to choose nodes to open (for pMILP,  but also somehow branching nodes for BaB). We agree that this is the main contribution of the paper. It is novel and is very promissing to bring more accuracy in the verification process.
>
> The verifier we obtained as byproduct (pMILP and then Hybrid MILP) should be seen as witness that the heuristic works extremely well, not as final perfectly rounded up verifier ready to be commercialized.
>
> There will always be new interesting hard networks, and we simply cannot test all of them.
>
> Similarly, to evaluate whether the new idea is promissing or not, I do not think it is necessary  to have implemented different activation functions. The class of ReLU DNNs is important enough and has enough examples to serve as benchmarks, and is thus sufficient for such an evaluation.
>
> We do not deny the interest of other activation functions, but we clearly put them as out of the scope of our paper (please wait for a journal version or a follow up paper).
>
>
> > I noticed that Table 4 in the appendix includes nine models, yet it seems Hybrid MILP was only applied to one of them. Was this decision entirely based on the #undecided images?
>
> Out of the 9 (13 with -4 variations) networks of Table 4 you are looking at, 7 (11) are "easy", as alpha beta Crown have very few undecided images %. As we explained, we focus on hard instances, as easy ones can be considered as "solved".
>
> The other 2:
> -CNN-B-Adv from the original abCROWN show an important number of undecided images, we consider.
>
> -resnet_4b_bn has abCROWN verifies 32% of images, with 41% undecided images (upper bound is at 73%).
> Resnet_4b_bn has 14.4K activation neurons. This was not in the original abCrown paper, reason why we didnt consider it (but again, there are probably 100 other hard DNN interesting to consider we didnt also considered, because we cannot consider all of them).
>
> (resnet_2b has abCROWN verifies 67% of images, with 0% undecided images. So not interesting)
>
> We commented on this DNNs Resnet_4b for the last reviewer (zhoq27).
> It seems believable that we can improve accuracy numbers here, as the number of activation function is <20k, less than CNN-B-Adv.
>
> Now, we are limited today by the fact that the resnet architecture is slightly different than CNNs or FCNNs, with residual blocks.
> The case is similar as MaxPool actually (your other comment):
>
> We would have to adapt the heuristic for residual layers, and/or for MaxPool activation function. This is not very hard to do, but it takes time. As I argued above, I do not think this is necessary in order to evaluate whether the new heuristic is promissing and deserves to be published.
>
> On Renet_4b: we actually experimented on it last week, and we could improve the accuracy over abCROWN by few %.
> However, we are severly limited by our heuristic which *today* does not take into account the residual layer, and it falls back to a much less accurate heuristic on almost all the layers. Adapting the heuristic is conceptually not difficult, but it takes some time.

---

### Official Review · Reviewer_BDUS · 2024-11-04

**Soundness:** 2
**Presentation:** 2
**Contribution:** 1
**Rating:** 5
**Confidence:** 4

**Summary:**

The authors propose a new algorithm for verifying neural networks against local robustness properties. This method takes a hybrid approach, combining an existing “Branch and Bound” verifier ($\alpha,\beta$-CROWN) with MILP-based verifiers. The authors show that this hybrid approach effectively reduces the number of undecided verification instances for non-robustly trained networks, where verification is known to be hard.

**Strengths:**

- Apart from a few typos, the paper is clearly written.
- The proposed approach effectively reduces undecided verification instances for non-robustly trained networks within a reasonable time limit.

**Weaknesses:**

**Motivation of the work**

Q1. This work focuses on networks that are challenging to verify because they are not robustly trained. Typically, standard-trained networks have low verified accuracy and are vulnerable to attacks. Given that these networks lack robustness, why should we even attempt to verify such difficult cases? For example, in recent papers on certified training [1], networks trained on MNIST and CIFAR-10 show verified accuracies higher than the upper bound in Table 1. Therefore, no sound verifier can achieve verified accuracy beyond what is reported in [1] on these networks. Additionally, even for ReLU networks robustness verification is known to be NP-hard, meaning that in the worst case, complete verification will require exponential time with respect to the number of unsettled ReLU nodes (assuming $ P \neq NP $). So in theory there will always be these hard instances and this was the reason behind certifiable training that makes verification easier.

[1] “Certified Training: Small Boxes are All You Need”, ICLR, 2023.

**Missing related works**

(Lines 296 - 298) The use of two lower bounds (commonly known as the triangle relaxation) within an LP formulation is not new and has been applied in prior works [1] and more recently in [2].


**Technical Contributions:**

The main technical contributions of this work are not entirely clear to me.

- Q1: Proposition 1 appears to be a well-known result and is used in existing works [1, 2]. The authors should cite the relevant papers.

- Q2: The authors do not provide sufficient detail on the hybrid-MILP algorithm. Authors should include a pseudo-code outlining the key steps of the proposed algorithm.

-  Q3: The high-level idea behind the utility function described in Section 4 closely resembles branching heuristics, such as BaBSR (cited by the authors), in terms of evaluating the importance of an unsettled neuron with respect to a specific verification property. The authors should clarify how their approach differs from existing branching heuristics.


[1] “Formal Verification of Piece-Wise Linear Feed-Forward Neural Networks”, ATVA, 2017.\
[2] “Input-Relational Verification of Deep Neural Networks”, PLDI, 2024.

**Missing Experiments with current SOTA verifiers**

$\alpha,\beta$-CROWN is no longer the SOTA verifier for local robustness. The authors should consider providing comparisons with GCP-CROWN and if possible with MN-BaB (on all networks table 4).

**Missing details in the experimental setup**

Q1. The authors should provide the config files (example - https://github.com/Verified-Intelligence/alpha-beta-CROWN/blob/main/complete_verifier/exp_configs/beta_crown/cifar_resnet_2b.yaml) they are using for comparing with $\alpha,\beta$-CROWN including the following details
- Branching heuristic
- Batch size
- Cuts are applied or not (as done in GCP-CROWN)

**UAP/Hyperproperties/Relational property verification:**

Is the proposed approach applicable to verifying robustness against UAP or more general hyperproperties? It appears that prior works [1,2,3] take a similar approach, combining abstract interpretation or bounding techniques with a MILP formulation that is easy to optimize.

[1] “Towards Robustness Certification Against Universal Perturbations”, ICLR, 2023. \
[2] “Input-Relational Verification of Deep Neural Networks”, PLDI, 2024. \
[3] “Relational DNN Verification With Cross Executional Bound Refinement”, ICML, 2024.



**Typos and Minor comments**

- Line 77 - Missing citation
- Line 174 - incomplete sentence
- Line 233 - “Box abstraction” - Interval or Box domain is a well-known abstract domain used in traditional program analysis and not introduced by the cited paper. The cited paper introduced DeepPoly a type of symbolic interval domain or restricted polyhedra domain.
- Line 310 - Typo `nappears`

**Questions:**

Refer to the Weaknesses section

---

> ### Author Response · Authors · 2024-11-12
>
> Thanks a lot for all the pointers.
> We are going to consider all of them, in particular run test on GCP-CROWN  on at least some of the benchmarks.
>
> However, to run our independant experiments, IBM CPLEX solver is needed to run GCP-CROWN.
> We just required the permission to use IBM CPLEX solver and waiting for the approval. We will run experiements when its done.
>
> What we can say for now:
> from [General Cutting Planes for Bound-Propagation-Based Neural Network Verification, NeurIPS 2022]:
>
> On CNN-B-ADV, GCP-CROWN is 2% more accurate than alpha,beta CROWN while taking 50% more time.
> At least on this important benchmark, this will not change the conclusion tremendously: we have 11% of undecided images vs 20-22% for alpha beta CROWN depending on the time-out duration. Our improvement is much larger than the 2% between alpha,beta CROWN and GCP-CROWN.
>
>
> On Parameters used when running abCROWN:
> The networks tested have already been tested by abCROWN.
> We thus use the configuration files which come for each network in abCROWN (except for time-out which we explicitly mention).
>
> e.g.: for CNN-B-Adv:
> https://github.com/Verified-Intelligence/alpha-beta-CROWN/blob/main/complete_verifier/exp_configs/beta_crown/cifar_cnn_b_adv.yaml
> "solver:
>   batch_size: 512
>   beta-crown:
>     iteration: 20"
>
> and
>
> or for 6x100:
> https://github.com/Verified-Intelligence/alpha-beta-CROWN/blob/main/complete_verifier/exp_configs/beta_crown/mnist_6_100.yaml
>
> "solver:
>   batch_size: 1024
>   beta-crown:
>     iteration: 20"
>
> Our experimental results are in line with published numbers.
> So far we did not use GCP-CROWN nor cuts.
>
>
>
> Q1:  Two important things.
> 1) "This work focuses on networks that are challenging to verify because they are not robustly trained".
> This is factually incorrect. Consider for instance CNN-B-Adv. It has been trained to be robust against Adversarial attacks.
> Also, we spent quite some time in the introduction to explain that while training to be robust and/or to be more easily verifiable is indeed possible, this can be done only up to a point before hurting accuracy, which is worst than being non robust.
> We certaintly agree that  for easy instances, alpha, beta Crown is very good, this is our point indeed. Then you have the not so easy (but not complexity worst case) instances where alpha, beta CROWN (and every other verifier we tested) does not work that well. And that is our focus (as the easy case can be considered "solved").  Please read the paragraph in the introduction starting at line 052 and ending by "one cannot expect only
> easy verification instances: hard verification instances need to be explored as well".
> Similar argument is made in the excellent Dathathri et al. (2020).
>
> 2) The fact that there are indeed some (rare) truely hard instances is not an excuse for alpha,beta-CROWN (and every other current verifiers we tested) for performing so poorly on so many instances. These instances where alpha,beta CROWN fails are also not "hard" instances in terms of worst case complexity, since our prototype succeeds into solving them in reasonable time (and our prototype does nothing magical - it is just carefully designed). This is a proof that alpha,beta Crown has some important gaps we try to solve (which is the case of MN-Bab also obviously ). We will test to see if it is the case for GCP-CROWN as well, and report on the results.
>
> Concerning the pseudo code:
>
> HybridMILP pseudo-code:
>
>     Run Alpha-Beta Crown with small time out (e.g. 10s).
>
>     If image undecided then run pMILP.
>
> pMILP pseudo code:
>
> Input: Bounds [α_m, β_m] for input nodes m at layer 0 (input neighbourhood)
>
> Output: Bounds [α_n, β_n] for every output node n
>
> for layer k = 1 · · · K do
>
>        for neuron n in layer k do
>
>            Compute the set Z of the K most important nodes before n according to Utility(n)
>
>           Run partial MILP(Z as binary, nodes / Z as linear variables, with constraints every node m<n satisfy x_m \in [α_m, β_m]  )
>          to obtain bounds [α_n, β_n] for node n.
>
> return Bounds [α_n, β_n] for every output node n
>
>
>
> The complexity for pMILP is the following:
>
> O(N · MILP(N, K)), where N is the number of nodes of the DNN, and MILP(N, K) is the complexity of solving a MILP program with
> K binary varialbes and N-K linear variables. We have that We have that MILP(N, K) < 2^K LP(N), where LP(N) is the complexity of running LP, polynomial in N, thus the exponential is only on K.
>
> The key factor is thus to keep K as small as possible, which our Utility function enables (see Table 3).

---

> > ### Comment · Reviewer_BDUS · 2024-11-13
> >
> > Thanks for the prompt reply.
> >
> > **Motivation (minor concern)**
> > The main point behind the motivation question was that none of the networks' verified accuracies could match those of a certifiably robust, state-of-the-art (SOTA) model (even with a perfect verifier). For reference, "in recent papers on certified training [1], networks trained on MNIST and CIFAR-10 show verified accuracies higher than the upper bound in Table 1." Therefore, I wanted to understand the practical importance of solving these challenging instances for an ML practitioner, given that there are already better models with higher verified and, in many cases, standard accuracy [1].
> >
> > **Technical concern (major concern)**
> > The technical concerns have not been addressed. The authors should clearly highlight the key technical contributions, especially considering that the following contributions are already known in the verification community:
> > - Proposition 1 is established in prior works.
> > - Selectively encoding unsettled ReLU nodes with binary variables improves verification accuracy.
> > - Bound tightening of intermediate layer neurons (as shown in Figure 3 [A]) improves verification accuracy.
> > - Utility functions, in principle, appear similar to branching heuristics (e.g., BaBSR).
> >
> > [A] "Fast and Complete: Enabling Complete Neural Network Verification with Rapid and Massively Parallel Incomplete Verifiers",  ICLR, 2021.
> >
> > **Pseudo code and complexity**
> > Thanks for providing the pseudo-code. However, it is still incomplete and does not mention any hyperparameters or whether iterations of the inner loop are executed parallelly.
> > The worst-case complexity, $O(N \cdot MILP(N, K))$, is too high even for small networks. This approach is feasible only when the number of unsettled neurons is low. I strongly recommend that the authors provide a runtime analysis using a higher
> > $\epsilon$ values, for example, $\epsilon$ values say $\epsilon \in [0.03, 0.035, 0.04, 0.05, ...]$ for the MNIST $5 \times 100$ network (Table 1 row 1). This is necessary to understand the scalability of the algorithm.

---

> ### Author Response · Authors · 2024-11-13
>
> >Technical concern (major concern),
>
> let us investigate before providing an answer - we will.
>
> On other concerns:
>
> >Motivation (minor concern): "CIFAR-10 show verified accuracies higher than the upper bound in Table 1."
>
> EDIT: the following is due to a misunderstanding on the paper [1] (there were several reference [1], in the same ICLR 2023 conference)
> >This is factually incorrect.
>
> >In [1] , the trained to be certifiable for robustness CIFAR10 ResNet 4B certification results is around 54% with epsilon= 1/255,>(table 3 page 7 of https://openreview.net/pdf?id=7GEvPKxjtt).
>
> >The upper bound for CIFAR10 CNN-B-Adv is at 62%, which is higher than 54%, not lower as claimed.
> >Even more importantly, the upper bound is for A MUCH HARDER epsilon=2/255 (perturbation twice as high).
> Notice that with epsilon= 3/255 [1] reaches 0% of certification (epsilon= 2/255 is not reported in  [1], so direct comparison is not that easy).
>
> >Even better, Hybrid MILP actually succeeds to certify 51% certified robustness for CIFAR 10 CNN-B-Adv for the MUCH HARDER epsilon= *2*/255, with perturbations twice as large than epsilon= 1/255. This is a much stronger result than the
> 54% of certification with epsilon= 1/255, and this can be cerified thanks to Hybrid MILP.
>
> This still stands:
> More generally, in academic world, we can train our own networks, but in the real world, you have to do with the network trained by the industrial partner, and the training pipeline often cannot be changed. This is also what is advocated in Dathathri et al. (2020). As we said, you cannot expect all the networks to be specifically trained to be easy to verify: some will be, some will not be - and we need new methods to tackle them all.
>
> What we did is that we experimented on standard benchmarks of the community.
>
> > Pseudo code and complexity: "The worst-case complexity, is too high even for small networks. This approach is feasible only when the number of unsettled neurons is low."
>
> Factually, the experimental results point to the exact opposite direction :) :
>
> pMILP efficiently certifies CNN-B-Adv, which has 20, 000 neurons and quite a lot of unsettled neurons (otherwise, abCROWN would have very few undecided images).
>
>
> pMILP is computing bounds for 20 neurons in parallel (as indicated line 689, page 13), very similar to PRIMA or Refined bCROWN (16 neurons in parallel), and much less than using >4000 GPU core in standard abCROWN.
>
> Several things to consider:
> 1) The complexity  allows to understand what is the crucial parameter to keep in track. Here, it is the number K of open nodes. And this is exactly what our main contribution does: identifying few very important nodes. If K is small, then the complexity is manageable efficiently even for large network (complexity polynomial in N).
>
> 2) worst-case complexity IS in the worst case. The average case is usually behaving better. We use the Gurobi solver, which has powerful heuristic which avoids the worst case complexity most of the time. That being said, our prototype is much more efficient than any previous verifier primarily using Gurobi, so this sole factor is only a small part of the reason.
>
> >  I strongly recommend that the authors provide a runtime analysis using a higher  values, for example,
>  values say  for the MNIST  network (Table 1 row 1), using higher  epsilon values. This is necessary to understand the scalability of the algorithm.
>
> You should look at figure 2 page 10: we certified some instances with epsilon as high as 0.1, for the complex 6x500 networks.
> Obviously, scaling to larger epsilon is not an issue. Actually, runtime does not depend directly on epsilon. It depends on the distance to the frontier between certifiable and falsifiable. asking for epsilon >> best falsifiable epsilon would return very fast a (negative) answer, similarly if epsilon << best certifiable epsilon.

---

> ### Comment · Reviewer_BDUS · 2024-11-13
>
> > In [1], the trained to be certifiable for robustness CIFAR10 ResNet 4B certification results is around 54% with epsilon= 1/255,(table 3 page 7 of https://openreview.net/pdf?id=7GEvPKxjtt).
>
> It seems there might have been a misunderstanding. I would kindly suggest that the authors review the cited papers more carefully before responding. Specifically, I referred to the paper "Certified Training: Small Boxes are All You Need" (https://openreview.net/forum?id=7oFuxtJtUMH, see Table 1). The certified accuracy at $2/255$ is reported as $62.84$%, which is higher than the upper bound of $62.0$% mentioned by the authors. That said, I do agree that there are certain benchmarks (networks) and specific properties (such as $\epsilon$ values) where using a MILP formulation can indeed help reduce the number of unverified instances.
>
> > Figure 2 page 10
>
> I may have overlooked the runtime details. Could you please point me to or provide specifics on the runtime, such as the number of MILP calls, average and maximum time per MILP call, and the hyperparameter values used? Additionally, could you share the same details for the other networks presented in Table 1? Furthermore, is there a particular reason why the authors did not consider Refined $\beta$-CROWN/ERAN for this experiment, as it also utilizes an LP/MILP-based approach?
>
> > $\epsilon$ values
>
> "It depends on the distance to the frontier between certifiable and falsifiable. asking for epsilon >> best falsifiable epsilon would return very fast a (negative) answer, similarly if epsilon << best certifiable epsilon." - I completely agree with this. However, I wanted to see the runtime details (e.g., MILP calls, hyperparameters, etc.) for different $\epsilon$ values in the range [best certifiable epsilon, best falsifiable epsilon]. Specifically, I am interested in the $\epsilon$ values close to the best falsifiable epsilon, but the network cannot be attacked.

---

> > ### Comment · Reviewer_ZYJY · 2024-12-02
> > **Comparing [1] and Hybrid MILP may not be reasonable.**
> >
> > Hey Reviewer BDUS,
> >  After reviewing [1], I believe that comparing its results with Hybrid MILP is not appropriate, as the two methods have fundamentally different goals. [1] improves model robustness through certified training, modifying the model’s weights to achieve higher verified accuracy. In contrast, Hybrid MILP focuses on improving the accuracy of robustness verification for already trained models, regardless of whether they were robustly trained or not, without altering the model itself. These approaches address different challenges. Could you kindly explain why such a comparison is being made?

---

> > > ### Comment · Reviewer_BDUS · 2024-12-03
> > >
> > > Dear Reviewer ZYJY,
> > >
> > > I did not ask the authors to compare their work with a certified training method [1]. Instead, I questioned the choice of networks used in this study, as these networks generally demonstrate lower certified accuracy. Additionally, the upper bound on robust accuracy for these networks is lower than that of the certifiably robust DNNs trained in [1].
> > >
> > > My concern was why verifying such models (or any models not trained for certified robustness) was considered an interesting problem, given that even with a perfect verifier (assuming sound verification), these DNNs would not achieve state-of-the-art robust accuracy.
> > >
> > > That said, I am satisfied with the authors' explanation and have decided to increase my grades accordingly.

---

> ### Author Response · Authors · 2024-11-14
>
> First I want to apologize for the misunderstanding on cited paper [1]. The issue was there were several [1] used in your original review (I was not aware of that), and I mistakenly took the wrong one (which also train networks and certifiy them, so it was easy to believe it was the one you were refering to, both at ICLR 2023). Anyway, our point is that there are many networks out there, we cannot test them all, and in real life you cannot necessarily choose the easiest one learnt for a specific task.
>
> We choose to experiment with standard DNNs already tested with abCROWN such that we can compare and use already mature configuration files.
>
> > the number of MILP calls
>
> This is 2 (one for lower bound, one for upper bound) per neurons, thus the O(N), for all DNNs per image per epsilon.
> For Figure 2, we have time to query ~6 different epsilon in 10 000s, in a binary search.
>
> > average and maximum time per MILP call
>
> We did not keep the statistic you are asking for.
> For average, it is easy to approximate accurately:
> average run time per  image x 20 (threadhs in parallel) / [ 2 (min and max) x Number of nodes].
> This gives:
>
> 5x 100: 0.92s
>
> 5x 200: 0.71s
>
> 9x 100: 0.76s
>
> 9x 200: 0.49s
>
> 6x 500: 1.34s
>
> CNN-B-Adv: 0.8s
>
> maximum is a lot higher. In particular for output nodes which have a higher number of open nodes, in particular for CNN-B-Adv.
> We also rerun pMILP on 1 image of each case:
>
> CIFAR on image 76, hard image as abCROWN  cannot certify with 2000s time out (gap to certification 0.25)
>
> CNN-B-Adv, epsilon = 2/255:  average 1s per MILP calls.
> max 609s (that is for the hardest output neuron - hence the number of open nodes is large, this is an extreme outlier case).
>
> We are running another easier image (that abCROWN certifies) for comparison.
>
>
>
> MNIST: For image 3, results are:
>
> 6x 500, epsilon=0.035 (representative of average case): average 1.4s per MILPcall, max: 3.5s.
>
>
> 6x 500, epsilon=0.1, very close to the falsification epsilon (very hard, not representative of the average case): avg: 44.6s, max: 310s
>
> 5x 100, epsilon=0.026: avg: 0.41      max: 1.87
>
> 5x 200, epsilon=0.015: avg: 0.75       max 5.31
>
> 8x 100, epsilon=0.026: avg 0.39	 max: 1.406
>
> 8x 200, epsilon=0.015: avg 0.49	max: 1.63
>
>
> > hyperparameters.
>
> If you mean MIP Gap or time out, we have our own ones. MIPGap varies from 0.001 to 0.1 depending on improvement, distance to deciding the neurons, etc.  This is low level details which will be available in the code on a Github after the paper is accepted.
>
> >Why didnt you try ith Refined $\beta$-CROWN/ERAN for this experiment
>
> Refer to line 467: Last but not least, Refined β-Crown cannot scale to larger instances (6×500, CNN-B-Adv), while Hybrid MILP can.
>
> Specifically, the engine return a 'not implemented' when we tried to run it. Anyway, running >500 nodes with full MILP would be extremely slow (already for the first layer using MILP - next layer wont probably run at all);
>
>  For Eran, refer to Table 6 page 14 in appendix, with results worse than abCROWN.
>
> > Specifically, I am interested in the $\epsilon$ values close to the best falsifiable epsilon, but the network cannot be attacked.
>
> See above the case of image 3 with epsilon=0.035 and epsilon=0.1 (much harder).
> Figure 2 shows that Hybrid MILP can certify images quite close to the boundary, much closer than abCROWN.
> On this image, pMILP succeds to verify epsilon= 1.054, while abCROWN can only certify epsilon = 0.0467

---

> ### Author Response · Authors · 2024-11-14
> **Proposition 1**
>
> > Major Concern: Proposition 1 is established in prior works.
>
> We are looking at each of the 4 papers mentioned by the reviewers on the issue.
>
> [A] “Formal Verification of Piece-Wise Linear Feed-Forward Neural Networks”, ATVA, 2017.
>
> [B] “Input-Relational Verification of Deep Neural Networks”, PLDI, 2024.
>
> [C] Singh et al. "An abstract domain for certifying neural networks" Equation 2.
>
> [D] Salman, et al. "A convex relaxation barrier to tight robustness verification of neural networks. 2019". Theorem 4.2
>
> We did not find a clear and explicit statement which would be equivalent with our Proposition 1. For papers [B] and [D], we are exploring further to see if their results (specifically Theorem 4.2) would imply Proposition 1 or if they would suggest in writing something similar to Proposition 1. For two papers: [A] and [C], the work are clearly orthogonal to our statement.
>
> We believe there might be a misunderstanding in what our statement claims, which would explain why [A] and [C] at least are suggested as establishing the results by the reviewers.
>
> To rephrase in English, Proposition 1 claims that the LP relaxation (of the standard exact MILP encoding of ReLU) is EQUIVALENT with the triangle approximation. And [A;C] are simply NOT dealing with LP relaxation!
>
> To say differently, Proposition 1 does not claim the triangle approximation to be novel – it certainly isn’t ([A],[B],[C],[D] are non exhaustive examples of that). But the EQUIVALENCE may be new, and at least it is not that obvious or well-known. That being said, it might already be proved in another paper ([D] or somewhere else). The proof is not that complicated, but this is certainly very important and instrumental in the way our Utility function is so much more efficient than what was proposed before (more on the utility function later). If Proposition 1 is already proved somewhere else, then it does not impact our paper much.
>
> We certainly agree that the idea of a triangle approximation appeared countless time before (in these 4 papers specifically, but in numerous other as well). We actually note such a fact immediately after Proposition 1, line 296 “Notice that this is close to the DeepPoly abstractions Singh et al. (2019b)…”

---

> ### Author Response · Authors · 2024-11-14
> **Utility Function vs Branching Heuristics.**
>
> > Major Concern: Utility functions, in principle, appear similar to branching heuristics (e.g., BaBSR).
>
>
> After investigating several papers, we can say the following:
>
> Most (if not all) heuristics in DNN verification consider how each (uncertain) neuron affects a certain node X (e.g. an output neuron X): e.g. BaBSR, Huang et al. 2020, etc.
>
> Our Utility function is no different in that respect: it ranks neurons by how important they are to a given node X, to only consider the most important neurons accurately.
>
> Similarities stop there with BaBSR and previous heuristics.
>
> Now, the key difference, and the reason why our Utility is so efficient (it can even make the usually very inefficient MILP solvers more efficient than the otherwise state of the art BaB solvers. Compared to Huang et al. 2020, it needs to open 4 times less neurons for the same accuracy) is the following.
>
> It uses the solution to an LP call of a min/max query on node X (as computed by Gurobi) (line 312). As it is LP, the solution computed is also the bound in the dual space. The cost of this LP call is negligeable wrt the forthcoming call to pMILP, and thus the time taken here is certainly worth.
>
> Now, we use this solution, which associates each variable/neuron with a specific value, to have a much more accurate understanding of how each variable/neuron affects X. That’s where Proposition 1 is crucial, in telling us how much accuracy is gained by opening that neuron vs LP relaxation, and we can rank them extremely accurately (Table 3 is extremely clear in that respect). Proposition 2 is interesting, but is far from capturing how close UTILITY is to IMPROVEMENT (in our test, we were 99% close- of course this cannot be proven in general as we could manually generate degenerated cases where the distance is much larger).
>
> Compared to e.g. Huang et al 2020, it allows to have a very precise answer *locally* for that input, while Huang et al. considers mostly the DNN, and less the input. This is also what happens with BaBSR, although comparing pMILP and BaB cannot be made very directly.
>
> To the best of our knowledge, this is the first time such a solution (through an LP call) is used to refine the contribution of each neuron to a given node X.

---

> > ### Comment · Reviewer_BDUS · 2024-11-24
> >
> > I thank the authors for their detailed response and the additional experimental evidence. I think the experimental results are more convincing now and I have decided to increase my grades.
> >
> > **Technical Novelty:** I still believe that Proposition 1 is not novel and is well-known from previous works. The triangle relaxation is defined as a conjunction of linear inequalities, as demonstrated in widely-cited methods like DeepPoly [1], which optimizes efficiency by greedily discarding one of the lower bounds during backpropagation to avoid invoking LP solvers (see page 7 in [1]). Alternatively, if LP solvers are employed, all linear constraints can be retained, as shown in [2] (see Eq. 9 and the discussion on page 15) and other related works. I strongly recommend the reviewers not to present Proposition 1 as a new contribution and instead rephrase it with appropriate references to prior work.
> >
> > [1] "An abstract domain for certifying neural networks", POPL, 2019. \
> > [2] “Input-Relational Verification of Deep Neural Networks”, PLDI, 2024.
> >
> > **Hyperparameters:** I was not specifically inquiring about Gurobi-specific parameters (although they are important). However, the value of $|X|$ appears to play a crucial role in balancing performance and runtime. Line 813 mentions that $|X| \in [21, 24]$ yields the best performance. I have a few questions:
> >
> > - How were these values determined by the authors?
> > - Were they identified by running multiple experiments and observing where the performance peaked?
> > - Does this range vary depending on the network, property (e.g., images, $\epsilon$ value), or both?
> > - If so, for the usability of the proposed tool, could the authors clarify how much manual effort and time were required to tune these hyperparameters? Should the time spent on hyperparameter tuning be accounted for in the experimental evaluation?
> > - Finally, does refined-$\beta$-CROWN using $|X| \in [21, 24]$ encounter runtime issues, as described earlier?

---

> ### Author Response · Authors · 2024-11-25
>
> >	Proposition 1.
>
> edit: we have pushed a new version of the pdf, with changes in the abstract, introduction (main contributions) and chapter 3.
> We believe the changes will  provide a consensual picture of the matter on Proposition 1.
>
> >	How were these values determined by the authors?
>
> This is an important question.
>
> For Fully Connected DNNs: Notice first that all of them use the exact same range for a given internal layer. The range is of size 3. Also, apart from the first and last layer (see below), the formula is min layer _ n+1 = [min layer_n /2]_rounded up. The first layer is a bit different: we started with min =43 (formula respected), and witnessed that adding 5 nodes was not impacting the runtime, whereas accuracy of the first layer was very important for the later layers, so we fixed it to 48. For the last layer, there is only 10 nodes and the accuracy is obviously extremely important, we set min=14. For choosing the exact number in the range, it is just based on Utilty function. We set a small threshold 0.01, and nodes are added from the minimum only if utility exceeds the threshold.
>
> Now, this gives few parameters: 21 nodes, /2, 14 nodes for last layer, threshold 0.01.
> We set these parameters by looking at one DNN (5x100) and one image (59), with the objective to spend roughly the same time in each layer (more variables in later layer means less binary variables were allowed to keep runtime reasonable). Few trial and error allowed us to set these parameters with around ten inferences in total.
> We tried on other images and FCNNs and it was working well. This was a bit surprising as the number of nodes per layer was quite different (from 100 to 500, with a variation of 1 to 6 times the number of linear variables in the MILP calls).
>
> Setting different parameters for different DNNs should improve the trade-offs (e.g. 5x100 could be made more accurate and slightly slower - it is 2 times faster than refined b-CROWN), but we didnt want to go that way.
>
> For CNN: the Convolutional architecture is quite different, and we had to have a different path.
> Again, we considered a single image to set up the parameters. For the first layer using pMILP, we witnessed that using X=200 was not too taxing on the runtime, while it was important for accuracy (similarly as for the FCNN). For the next layer (fully connected), we save time by using plain LP, as the number of variables is large and the number of neuron to consider in the layer is also large. It was a better trade off to increase the number of open nodes in the output layer, as the number of neurons to consider in the output layer is 10, again similar to what we saw for FCNN. For the last layer, we start with opening 45 neurons, and let the Utility function increase this number until solution >0.1. Setting a lower number hurts runtime because Gurobi needs to find a bound very close to the solution. Setting a higher number hurts the runtime because Gurobi as too many binary variables.
>
> Again, we set these few parameters (200, 45, >0.1) by few inferences and trial and error process on this one image. We then run on other images and it was working well.
>
> >	Were they identified by running multiple experiments and observing where the performance peaked?
>
> As explained above, no we did not run all images and all DNNs multiple time and cherry picked the best number. We would not have the computational power to do that anyway.
>
> >	Does this range vary depending on the network, property (e.g., images, � value), or both?
>
> It depends on the DNN, the image, and the neuron. We did not hard-code this number for each neuron. Instead, we let pMILP adapt according to the Utility value for nodes e.g. ranked 22, 23 and 24. It should be seen as a way to save a bit of runtime when Utility knows that the next node by ranking (e.g. 22) will not impact accuracy much (thanks to Proposition 2).
>
> >	If so, for the usability of the proposed tool, could the authors clarify how much manual effort and time were required to tune these hyperparameters? Should the time spent on hyperparameter tuning be accounted for in the experimental evaluation?
>
> We did not account for the parameter setting in Table 4. We believe it is standard practice to set these kind of parameters as part of the development process. This is negligible anyway compared to certifying many images, as this is a one-time cost, independent on the number of certified images.

---

> ### Author Response · Authors · 2024-11-25
>
> >	Finally, does refined-b-CROWN using |X|∈[21,24] encounter runtime issues, as described earlier?
>
> We are not totally sure of how refined-b-CROWN is implemented internally, but we believe it is calling iterative full MILP layer by layer on some of the nodes. If it is using reduced number of neurons as binary constraints, then it is obviously not as efficient as our Utility function judging by the results in Table 4, and also low level comparison of logs between refined-b-Crown and pMILP on same neurons. It does not seem possible to change the number of open nodes in the parameter file (we used the one which comes with refined beta-Crown for these networks). For larger DNNs, 6x500 and CNN-B-Adv, refined b-CROWN return a non-implemented function, which points to hard coded parameters in the code. If the number of binary variables can be set in the code, we believe the authors of refined b-CROWN have optimized this number already wrt the ranking function they are using. Changing only this parameter *alone* is likely not going to improve refined-b-crown.
>
> That being said, a version of refined-b-CROWN using our Utility function and partial MILP should work much better than refined b-CROWN. That would require to change the code heavily though.

---

### Author Response · Authors · 2024-11-22
**new version of the PDF**

Dear Reviewers,

Thanks again for your very constructive reviews.
The discussion phase is ending soon. We would like to have your feedback on our revised paper:
We have produced a new pdf file including all of what was discussed, answering all of your concerns, with blue highlights of the major changes.


-Introduction : we rewrote Proposition 1 statement to make the novelty clearer.
We also explained the novelty of our Utility function (it is the first function to rank nodes using a call to a (LP) solver, which makes it particularly efficient, as shown in Table 3).


Section 3 introduce the triangular abstraction first (well known), then later state Proposition 1 (new), with a note that the statement is new up to our knowedlge (in particular, we cannot find it in the referred papers [A,B,C,D]).

[A] “Formal Verification of Piece-Wise Linear Feed-Forward Neural Networks”, ATVA, 2017.

[B] “Input-Relational Verification of Deep Neural Networks”, PLDI, 2024.

[C] Singh et al. "An abstract domain for certifying neural networks" Equation 2.

[D] Salman, et al. "A convex relaxation barrier to tight robustness verification of neural networks. 2019". Theorem 4.2


Section 4: we rewrote Section 4 entirely, stating the key novelty (the fact that we use a (single) call to LP solver in order to rank nodes, which is novel as far as we know, and makes the choice much more accurate than previous attempts), as well as provide a clearer and shorter definition of the Utility function.

Appendix A: has all the necessary parameters for both pMILP and alpha beta CROWN, plus a note on GCP-Crown.

Appendix B: provides the pseudo code and the complexity analysis.
We also added a comparison with other papers using to improve verification accuracy:
 >Selectively encoding unsettled ReLU nodes with binary variables, and

 >Bound tightening of intermediate layer neurons (as shown in Figure 3 [A])

explaining why our approach is more efficient.

In particular, it scales to larger networks than previous attempts using MILP calls.

Appendix C: We added the ablation studies with graphs, showing the efficiency of each features of pMILP.

Appendix D: We kept the comparison with verifiers other than abCROWN.

Appendix E: We added the average and Max time per MILP calls.

Best,

---

### Author Response · Authors · 2024-11-27
**Latest version of the PDF**

To all reviewers:
Thanks again for the constructive direction to explore.

We commited the latest version of our PDF, with a large rewrite around Proposition 1, and conceptual comparison between Utility and other heurstics, including Bab heuristics, and rework of graphs.

In particular, in page 16, we added an experiment using BaB heuristic (SR and FSB) to choose ReLU nodes to open.
This does not work that well, because the objective to find nodes to branch on, and the objective to open a ReLU node for MILP are not exactly aligned. But the result is interesting nevertheless.

That being said, we believe that novel techniques we are following to open MILP nodes could be potentially adapted to rank nodes to branch on for BaB.

**EDIT:** actually, even without any adaptation, it looks like our Utility function already outperforms FSB / SR  when used as ranking in alpha,beta CROWN (pure BaB) (see post/table above), although its objective is *not* optimized for BaB branching.

---

### Author Response · Authors · 2024-12-03
**Latest experiment and interesting results for BaB / alpha,beta CROWN 2/2**

We use DNN 5x100, image 59. We consider nodes in layer 3, that is, exactly the same setting as Figure 2, 5 and 6 as well as Table 6 and 7.

However, we use alpha,beta CROWN to compute the bounds, and compare FSB with Utility to rank BaB nodes to branch on. We checked that FSB is slightly better than SR using alpha,beta CROWN, although both are very close.

We precomputed the best possible achievable bounds using full MILP for this layer to set the verification goal: we set the goal to verify as these bounds. The verification is obviously not achievable, but then we can  check how close to the bound the different heuristics get for a fixed time-out:
TO means time out, and -X.XX is the latest bound achieved (closer to 0 means closer to verification). We write in bold the best result for each case.

 We fixed the time out at 30s, 300s for both FSB and Utility. In order to get verification, we also relaxed the objectives with bound +0.2, .. +0.5 till verified.

| Node, Objective ...  | ... TO = 30s ... | ... TO = 300s ... |
| --- | --- | --- |
| Node0,  |  FSB:  TO, -1.0819  | FSB:  TO, -1.0819  |
| UB < -0.05643  |  Utility: TO, **-1.0492** |  Utility: TO, **-1.0492** |
| Node0,  |    |   FSB:  TO, -0.6819|
| UB < 0.343562  |   | Utility: TO, **-0.6492** |
| Node0,  |    |   FSB:  verified in 253s|
| UB < 0.443562  |  |    Utility: verified in **172s**|
| --- | --- | --- |
| Node1,  |  FSB: TO, -0.5927  | FSB: TO, -0.5927   |
| UB < 0.08124   |  Utility: TO, **-0.5726**       | Utility: TO, **-0.5726**  |
| Node1,  |    |  FSB: TO, -0.3092  |
| UB < 0.28124   |  |  **Utility: verified in 92s**  |
| --- | --- | --- |
| Node 2,  |  FSB: TO, **-0.9094**  |   FSB: TO, -0.9094 |
| UB < -0.0964   |  Utility: TO, -1.2575 |  Utility: TO, **-0.7785**  |
| Node 2,  |    | FSB: TO, -0.5094   |
| UB < 0.3036   | | **Utility: verified, 66s** |
| --- | --- | --- |
| Node 3,  |  FSB: TO, -0.7312| FSB: TO, -0.7312   |
| UB < 0.10694   |  Utility: TO, **-0.6926**     | Utility: TO, **-0.6926**  |
| Node 3,  |  | FSB: TO, -0.0.4271  |
| UB < 0.40694   |      | **Utility: verified, 112s**  |
| --- | --- | --- |
| Node 4,  |  FSB: TO, -0.6979  | FSB: TO, -0.6979   |
| UB <  0.22181  |  Utility: TO, **-0.6735**  |  Utility: TO, **-0.6735**  |
| Node 4,  |   | FSB: TO, -0.4611   |
| UB <  0.42181  |   |  **Utility: verified, 233s**  |
| --- | --- | --- |
| Node 5,  |  FSB: TO, -0.6474  |  FSB: TO, -0.6474  |
| UB <  0.59545  |  Utility: TO, **-0.6345** |  Utility: TO, **-0.6345**  |
| Node 5,  |   |  FSB: verified, 156s  |
| UB <  0.89545  |   |  Utility: verified, **111s**  |
| --- | --- | --- |
| Node 6,  |  FSB: TO, -0.6682   |  FSB: TO, -0.6682  |
| UB < 0.24122  |  Utility: TO, **-0.6359** | Utility: TO, **-0.6359**  |
| Node 6,  |  |  FSB: verified, 217s  |
| UB < 0.54122  |  | Utility: verified, **125s**  |
| --- | --- | --- |
| Node 7,  |  FSB: TO, - 0.8094 | FSB: TO, - 0.8094   |
| UB < 0.4426  |  Utility: TO, **-0.7248**        | Utility: TO, **-0.7248**  |
| Node 7,  |  | FSB: TO, - 0.0105   |
| UB < 0.7426  |      | **Utility: verified, 119s**  |
| --- | --- | --- |
| Node 8,  |  FSB: TO, -0.6069  |   FSB: TO, -0.6069 |
| UB < 0.09423  |  Utility: TO, **- 0.5983** | Utility: TO, **- 0.5983**  |
| Node 8,  |   |   FSB: TO, -0.4069 |
| UB < 0.29423  |  | **Utility: verified, 148s**  |
| --- | --- | --- |

Overall, our utility function to rank nodes to branch on for BaB in alpha,beta CROWN is significantly more accurate (or faster) than when using FSB with the same time out, except for node 2, Time Out 30s. Using TO=300s does not improve except for this  node 2, Utility becomes better than FSB. Also, on the same node 2 it succeeds to verify bound + 0.4 in 66s while FSB fails after timing out at 300s, with a latest bound far from verification (-0.5).

The reason of the discrepency for node 2, TO=30s is because it takes 3s to compute our Utility function. We tested with TO=33s for node 2, and we obtain: Utility=-0.7785 while FSB remains at -0.9094, confirming the 3s delta.
We add another test on simpler verification objectives, for bound + 0.4.

| Node, Objective | FSB | Utility|
| --- | --- | --- |
| Node 1, UB < 042812 | **17s** |  19s  |
| Node 2, UB < 0.303   | TO 300s  | **81s** |
| Node 3, UB < 0.5069 | 91s |  **40s**  |
| Node 4, UB <  0.6218   | 24s  | **23s** |
| Node 5, UB <  0.9954  | 32s |  **32s**  |
| Node 6, UB < 0.6412  | 42s  | **28s**  |
| Node 7, UB < 0.8426 | 55s | **47s**  |
| Node 8, UB < 0.4942  | **17s** | 20s  |
| average | >72s | **36s** |

For some nodes (1,8), FSB is 2-3s faster.
Overall, Utility is >2x faster.

Our Utility objective is unchanged, optimizing the difference (MILP – LP, using the solution of *one* LP solver call) to pick nodes to open for pMILP, without (yet) adaptation to optimize for BaB. So this is very promissing to obtain a very efficient heuristic for BaB as well, thanks to the novel ideas in our paper.

---

### Author Response · Authors · 2024-12-04
**Latest experiment and interesting results for BaB / alpha,beta CROWN. 1/2**

EDIT: We have to break up this comment in 2 as the Table is taking too much space.

We have finished evaluating nodes of layer 3 using pure BaB (in alpha beta Crown), comparing our Utility function with the FSB heuristic to rank nodes to branch on.

We provide here the interesting experiments with TO=300s, with tightest bound for which at least one of FSB / Utility verifies. See next post for the full table.

Alpha Beta Crown with verification for nodes of layer 3 of 5x100, image 59, time out 300, using FSB or Utility (formula unchanged from our paper) to rank nodes to branch on using pure BaB.

| Node, Objective | FSB | Utility (our) |
| --- | --- | --- |
| Node 0,  UB < 0.443562 | verified in 253s | verified in **172s** |
| Node 1, UB < 0.28124 |  Time Out 300s |  **verified in 92s**  |
| Node 2, UB < 0.3036   |    Time Out 300s  | **verified in 66s** |
| Node 3, UB < 0.40694 |  Time Out 300s  |  **verified in 112s**  |
| Node 4, UB <  0.42181   |   Time Out 300s  | **verified in 233s** |
| Node 5, UB <  0.89545  |     verified in 156s  |  verified in **111s**  |
| Node 6, UB < 0.54122  |  verified in 217s  | verified in **125s**  |
| Node 7, UB < 0.7426 |   Time Out 300s        | **verified in 119s**  |
| Node 8, UB < 0.29423  |    Time Out 300s | **verified in 148s**  |

Utility is more accurate than FSB for every node, reaching verification when either FSB cannot conclude and Times-out at 300s (node 1,2,3,4,7,8, average time for Utility 128s) or FSB runs 50% slower (nodes 0,5,6, average time 136s for Utility vs 209s for FSB). Over all nodes, Utility needs ~130s in average to reach verification (9/9 objectives), while FSB needs >>270s in average (6/9 nodes are still not verified!). A >>2x speed-up by just plugging in our Utility function unchanged from our paper into alpha,beta CROWN.

There is one small caveat, which is Utility takes 3s more to compute the ranking than FSB (integrated in the reported runtime). This explains why node 2 Time-Out 30s was more accurate for FSB than Utility (see table next post), because Utility needed 3 more seconds for the same number of computations.  With 33s, Utility is more accurate than FSB with 33s. Overall, this runtime overhead is very limited, almost negligible.

Notice that our Utility objective is unchanged, optimizing the difference (MILP – LP, using the solution of one LP solver call) to pick nodes to open for pMILP, without (yet) adaptation to optimize for BaB. **So this is very promissing to obtain a very efficient heuristic for BaB as well, validating further the novel ideas in our paper.**


Original post:

Dear Reviewers,

We hope that this latest experiment will further convince you to recommend acceptance, if you agree that our Utility function is both novel and very accurate. We strongly believe that the reasoning and the experiments in the paper prove that without any doubt (Figure 2 page 8 and fig 5 page16), confirmed by the accuracy results of Hybird MILP (Table 3 page 9). This latest experiment now provides even more promising research direction for the Utility function, and it validates further our novel strategy.

Thanks again for the constructive comments: we didn’t think of comparing our Utility function with BaB branching heuristics because the objectives were not the same.

Again, using FSB or SR branching heuristic to choose nodes to open for pMILP was not good (see page 16 in appendix, with FSB slighty worse than SR worse than Huang et al much worse than Utility). Now, that got us curious how our Utility function was fairing as heuristic to rank nodes to branch on with BaB. For that:

1/2

---

### Author Response · Authors · 2024-12-04
**Final Wrap Up**

Final Wrap up.

For the last time, we would like to thanks the reviewers for their constructive direction.

We believe we have tremendously improved the paper since submission.

The discussions revolved around our Utility heuristic, why is it so efficient, and compare it with more heuristics, in particular the ones used to rank nodes to branch on for BaB.

We rewrote the paper focusing on explaining conceptually how different and novel our Utility function, in particular using a primal solution to rank nodes, as well as better and stronger comparison with other heuristics. The final figure 2 (page 8) and 5 (page 16) strongly points to our Utility function being on another level of efficiency compared to the rest of the heuristics when used within a partial MILP framework.

The efficiency of our Utility function results in tangible improvements in accuracy and efficiency for verifying hard DNNs which could not be tackled by previous verifiers. While we agree that many more DNNs would be interesting to consider, we believe the spectre we investigated is sufficient to validate the efficiency of the heuristic, in particular with a quite large CNN-B-Adv with 20 000 activation, where 7% more images can be verified than the state of the art using similar runtime (accounting for GCP-CROWN being 2% more accurate than abCROWN on that CNN-B-Adv), with other networks (6x500) showing even stronger 40% improvement.

Out of curiosity, and because reviewers asked us to compared with BaB heuristic within partial MILP, we lately considered the converse: using our Utility function within a BaB framework (here, alpha, beta Crown), mimicking the setting of Figure 2 (for pMILP). Surprisingly, while the objective of our Utility function was *not* optimized for branching nodes in BaB but rather optimized for pMILP, this Utility function was outperforming heuristics for BaB (FSB and SR), more than twice as fast for the same accuracy (see the next two posts). While this is a restricted framework, this shows a lot of potential to apply our novel primal solution idea in this other context of BaB as well as in the context of pMILP we fully develop in the paper.

Some may wonder why we did not test Utility in alpha beta Crown to verify fully DNNs, and compare with alpha beta Crown using FSB. There are two reasons:

-	Technically, this would require to implement our Utility function ranging over all nodes, not only nodes from the 2 or 3 last layers. Such a function ranking nodes over all layers was not necessary with our pMILP strategy. With the little time remaining, we focused on obtaining results with already implemented tools, which gives another very interesting finding pointing to our methodology to use primal solution being also very promising in the BaB scope, not only for partial MILP.

-	Our paper is within the partial MILP scope, so such tests with BaB are not necessary. The point is mainly to showcase to the reviewers the potential impact of our novel techniques outside of the pMILP realm. We believe that the additional experiments we provided lately on layer 3 using BaB demonstrate just that. The bottom line is: More work is necessary to develop such an efficient and accurate heuristic based on primal solution to rank nodes for BaB, and this is way out of the scope of the paper.

---

### Meta-Review · Area_Chair_QpH1 · 2024-12-22

**Metareview:**

This paper studies the neural network verification problem to give guaranteed bounds on neural network outputs. It aims to improve the verification bound by selecting a certain set of important ReLU neurons that will be considered with integer variables in the MIP verification formulation, while other neurons use LP relaxation. A heuristic was proposed to select the set of neurons.

Reviewers rated the paper as a borderline paper. This paper improves upon the state-of-the-art verifier α,β-CROWN in a few settings, which is quite nice. The technical contribution of the paper, although not very significant, is valid and valuable. However, the writing of the paper is quite poor, especially the mathematical notations are not rigorous and often misleading (Section 4), and there are factual errors in the paper (see next paragraph). The quality is clearly below an average ICLR paper in this field. In addition, the scenarios considered in this paper are quite limited (although this is not a main concern for rejection), mostly fully connected MNIST models and very small CIFAR models.

Some claims in the paper are incorrect, e.g., branching heuristics are measuring "sensitivity" (in fact, what is "sensitivity" is unclear). Branching heuristics themselves are also developed to estimate the bound improvements after branching, exactly the same way as in this paper. For example, in the BaBSR paper, the heuristic was derived by calculating the difference between the two bounds, one with the neuron relaxed and one with the neuron fixed. A stronger estimation of bound improvement will lead to better branch-and-bound performance, so the findings during the rebuttal period were not a surprise. I suggest the authors to further explore the potential of their formulation as a branching heuristics. In addition, as also pointed out by Reviewer zhoq, the AC is very confident that Proposition 1 is a well known fact in this field and should not be claimed as a contribution.

Nevertheless, the feedback from the reviewers have greatly helped improve the paper and I encourage the authors to continue working on this subject and improving its quality.

**Additional Comments On Reviewer Discussion:**

The reviewers and the authors had multiple rounds of discussions during the rebuttal period and the authors provided many additional results. Some of the concerns from the reviewers were addressed, however there was no strong support from any of the reviewers and they believe the paper is on the borderline. The AC also read the paper carefully and believes the overall quality needs improvements for acceptance.

---

### Decision · Program_Chairs · 2025-01-22

Reject